# CaMiT: A Time-Aware Car Model Dataset for Classification and Generation

Frédéric Lin     Biruk Abere Ambaw     Adrian Popescu     Hejer Ammar
Romaric Audigier     Hervé Le Borgne

Université Paris-Saclay, CEA, List, F-91120, Palaiseau, France
`{firstname.lastname}@cea.fr`

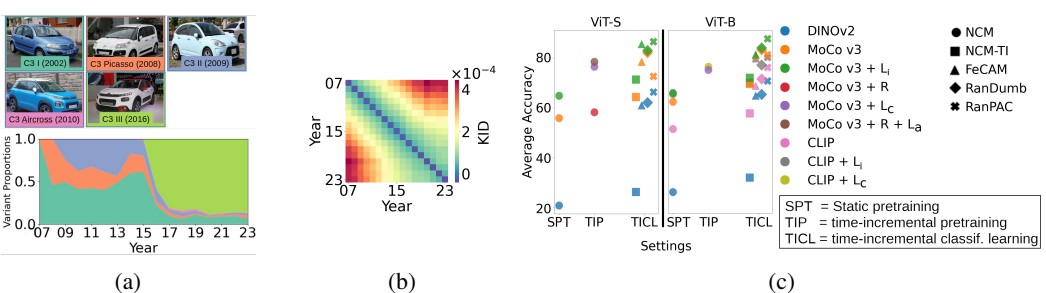

Figure 1: Illustration of CaMiT content and experiments. (a) exemplifies the *Citroën C3* data shift in time. (b) shows that the averaged kernel-inception distance between class embeddings increases with time. (c) highlights the positive effects of time-aware classification over static pretraining.

## Abstract

AI systems must adapt to the evolving visual landscape, especially in domains where object appearance shifts over time. While prior work on time-aware vision models has primarily addressed commonsense-level categories, we introduce Car Models in Time (CaMiT). This fine-grained dataset captures the temporal evolution of this representative subset of technological artifacts. CaMiT includes 787K labeled samples of 190 car models (2007–2023) and 5.1M unlabeled samples (2005–2023), supporting supervised and self-supervised learning. We show that static pretraining on in-domain data achieves competitive performance with large-scale generalist models, offering a more resource-efficient solution. However, accuracy degrades when testing a year's models backward and forward in time. To address this, we evaluate CaMiT in a time-incremental classification setting, a realistic continual learning scenario with emerging, evolving, and disappearing classes. We investigate two mitigation strategies: time-incremental pretraining, which updates the backbone model, and time-incremental classifier learning, which updates the final classification layer, with positive results in both cases. Finally, we introduce time-aware image generation by consistently using temporal metadata during training. Results indicate improved realism compared to standard generation. CaMiT provides a rich resource for exploring temporal adaptation in a fine-grained visual context for discriminative and generative AI systems.

## 1   Introduction

Large-scale visual datasets, such as ImageNet [9], LAION [58], or DataComp [15] play a central role in the development of AI systems. They are designed for a one-off training paradigm, not considering

39th Conference on Neural Information Processing Systems (NeurIPS 2025) Track on Datasets and Benchmarks.

novelty integration [7], and disregarding the appearance shift of visual classes over time. This shift is most prominent for technological artifacts resulting from changing designs [8, 11, 19], trends, and camera evolution [38, 47]. Existing work on temporal shift modeling focuses on time-continual pretraining of generic VLMs [16], coarse-grained visual class modeling [48], modeling specific artifact versions [15], or very specialized sensors such as satellite [5]. However, none of these datasets answers the following simple yet important question: *How should the depictions of fine-grained technological artifacts over a long period be modeled in visual models?*

To address this gap, we introduce CaMiT, a new time-aware dataset representing car models. We build on existing practices [23, 28, 71] to obtain raw samples from Flickr, the only public source allowing a collection over nearly 20 years. We propose a semi-automatic labeling pipeline combining VLMs and supervised models to obtain accurate labels while reducing manual effort. CaMiT includes a labeled subset describing 190 car models with 787K samples published between 2007 and 2023, and a pretraining dataset with 5.1M unlabeled samples spanning from 2005 to 2023. We illustrate the temporal shift of car model depictions for *Citroën C3* in Figure 1 (a). It shows the appearance of new variants and the decreasing importance of older ones toward the end of a car's lifespan (11 and 14 years [21, 43]). This shift leads to an increasing distance between car model embeddings when the gap between the photo uploads increases (Figure 1 (b)), and calls for temporal data shift mitigation.

We use CaMiT in three classification settings to evaluate the temporal data shift effects: (1) static pretraining (SPT) to analyze the effects of time without mitigation measures, (2) time-incremental pretraining (TIP) to understand the effect of updating models when new data become available, and (3) time-incremental classifier learning (TICL) to focus on the classification layer updates with frozen backbones. The results summarized in Figure 1 (c) show that: (1) specialized pretrained models, trained with CaMiT, have competitive performance compared with generic models in the SPT setting; (2) TIP partially mitigates temporal data shift, particularly when adapting pretrained models with LoRA using CaMiT labeled training data; (3) TICL provides even better accuracy, with the best performance obtained when gradually updating classifiers on top of a specialized pretrained backbone. These results show the importance of mitigating temporal effects and support the development of specialized models for fine-grained classification, alongside foundation models. We also introduce a time-aware image generation (TAIG) task to address the temporal data shift in generative models. We add temporal metadata in the training captions and show that this simple change makes the distribution of generated images more faithful to the actual one than conventional generation. We release the dataset https://huggingface.co/datasets/fredericlin/CaMiT and code https://github.com/lin-frederic/CaMiT to foster research on the understudied fine-grained temporal shift modeling.

## 2 Related work

**Fine-grained visual categorization (FGVC)** focuses on distinguishing specific visual categories [66, 64]. Datasets span human-made objects, natural categories, satellite imagery, or structured visual domains. General-purpose datasets such as ImageNet [9], LabelMe [56], and OpenImages [30] include car images without temporal metadata. Dedicated datasets like StanfordCars [28], Comp-Cars [71], VMMR [60], and Car-1000 [23] often include production years, but they lack explicit timestamps for images and do not capture the temporal evolution of model depictions. Also, they typically include under 150 samples per class, making them unsuitable for long-term analysis.

**Continual Learning (CL)** aims to integrate new data without forgetting prior knowledge [65]. Typical scenarios include task-, class-, and domain-incremental learning [63] (IL). In Task-IL, semantically disjoint datasets (e.g., Flower-102 [44], EuroSAT [20], Food-101 [3]) appear sequentially, and task identity is known at inference [73]. Class-IL splits general classification datasets (e.g., CIFAR-100 [29], ImageNet LSVRC [55], Google Landmarks [68]) into random subsets, without revealing the task label at test time [39]. Domain-IL assumes a fixed label space but introduces data distribution shifts over time (e.g., CORe50 [34], reordered ImageNet variants). More general cases address mixed-task and unknown-distribution settings [76]. Despite their widespread adoption, these setups overlook the natural temporal evolution of visual data that CL should account for.

**Several datasets model the temporal data shift in visual data.** Fine-grained examples include AmsterTime [72] (historic vs. modern street views), FMoW [5] (land use monitoring from satellite images), Yearbook [17] (portrait photos over decades), and AgeDB [42] (age progression in faces). However, these datasets are limited in scale and focused on specific domains. More generalist

Table 1: Comparison of temporally annotated visual datasets.

| Dataset | Yearbook | FMoW | AgeDB | AmsterTime | VCT-107 | TIC DataComp | **CaMiT** |
|---------|----------|------|-------|------------|---------|--------------|-----------|
| Input | Portraits | Satellite | Faces | Landmarks | Web images | Web images | **Web images** |
| Prediction | Gender | Land use | Face ID, Age, Gender | Place recognition | Object recognition | VLM pretraining | **Car recognition** |
| Years | 1930–2013 | 2002–2017 | ∼1890–2017 | ∼1850–2020 | 2007–2020 | 2014–2022 | **2005–2023** |
| #classes | 2 | 63 | 568 | 1.2K | 107 | – | **190** |
| #samples | 37K | 119K | 16K | 2.5K | 951K | 13B | **7.5M** |

datasets have recently emerged. TIC-DataComp [16] augments DataComp [15] with time metadata for continual pretraining of vision-language models, but it only provides sufficient coverage for the 2016–2022 period. CLEAR [32] and VCT-107 [48] capture temporal drift in 11 and 107 commonsense-level classes, respectively, using Flickr data. While TIC-DataComp targets broad multimodal representation learning, and CLEAR and VCT-107 focus on general visual concepts at a coarse semantic level, CaMiT concentrates on a single fine-grained category—cars—with dense year-level annotations and curated labels. This design enables controlled studies of gradual appearance evolution and long-term temporal modeling that are not feasible with existing datasets. Table 1 summarizes representative temporally annotated visual datasets in terms of their input modality, task definition, temporal coverage, number of classes, and overall scale.

**Dataset usage** is built on top of recent advancements in several visual tasks. Large-scale pretraining provides transferable features for downstream tasks, and we test them in zero-shot learning with our dataset. We experiment with representative existing training strategies: DINOv2 [46] and CLIP [51] for car model classification through time. Building on continual pretraining [7, 16], we compare them with static and dynamically updated domain-focused models. Then, we use recent continual image classification algorithms [24, 18, 40, 50] that leverage pretrained models to balance effectiveness and efficiency. Finally, inspired by works investigating the role of time in LLMs [6], continual image generation [74], and GAN-based portrait restoration [37], we investigate time-conditioned image generation in an attempt to make synthetic content more realistic.

## 3   Dataset

We introduce CaMiT to analyze the effect of time in fine-grained image classification and generation. We select car models due to their evolving designs [21] and the availability of sufficient data for these classes. We illustrate the dataset creation pipeline, including image collection, filtering, and annotation, in Figure 2. CaMiT comprises two subsets: a labeled subset with 787K samples of 190 car models (48 brands) and an unlabeled pretraining subset with 5.1M images. The brand/model distribution is: 27/99 European, 13/60 Asian and 8/31 American. The most represented brands are Ford, Audi, and Toyota, with 14, 10, and 10 models respectively. Appendix A provides further details, including visual model descriptions, hyperparameters, thresholds, and prompts. Finally, we analyze the content of CaMiT, emphasizing the temporal dynamics of car model representations.

**Data collection.** While building billion-scale image collections from publicly available sources is feasible [15, 58], constructing datasets with fine-grained labels and temporal metadata introduces distinct challenges. The following requirements guide our collection process: (1) the dataset must span a long period and include year-level temporal metadata, (2) it should cover a wide range of car models, (3) the images must depict exterior car views suitable for model recognition, and (4) the samples should present sufficient visual complexity to avoid performance saturation. The first requirement is specific to our work, while the latter three come from prior FGVC efforts [28, 71, 23]. Initially, we tried collecting data by testing Google Images' temporal filter, hoping to benefit from Google's extensive web image index. However, we found the temporal filter unreliable, with many images incorrectly timestamped. We also considered TIC-DataComp [16], a temporally enriched version of DataComp [15], which comprises nearly 13 billion images from 2014 to 2022. Despite its scale, this dataset exhibits significant temporal imbalance, with sufficient car model coverage only from 2017 onward, making it unsuitable for our needs. Consequently, we followed a more targeted approach using the Flickr API, consistent with established practices in dataset collection [31, 30, 48, 61]. We queried with car subtypes, brands, and models, applying Flickr's built-in temporal filter. We issued

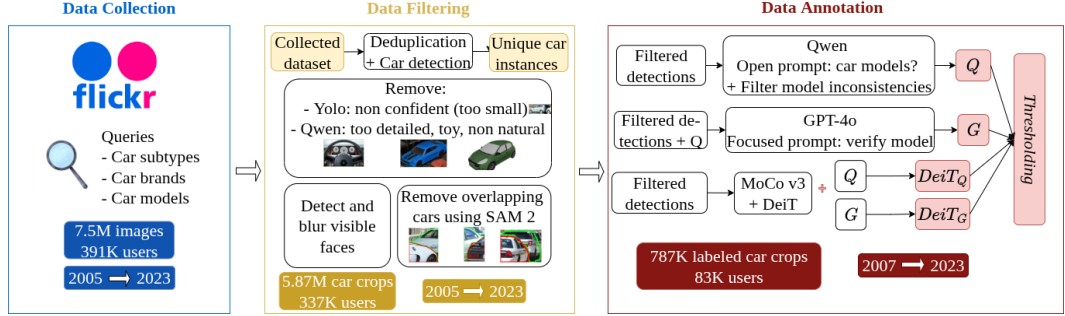

Figure 2: **Overview of the CaMiT curation pipeline.** The pipeline consists of three stages: (1) **Data Collection** — Flickr queries by car subtype, brand, and model yield 7.5 M candidate images. (2) **Data Filtering** — automated filtering and de-duplication remove non-relevant or low-quality samples, resulting in 5.87 M car crops. (3) **Data Annotation** — semi-automatic labeling with Qwen, GPT-4o, and DeiT produces 787 K verified car crops.

425 unique queries, collecting up to 5 000 images per query per year from 2005 to 2023. This process resulted in an initial dataset comprising 7.5 million images.

**Data filtering** removes duplicate, irrelevant, or low-quality samples, such as those lacking visible cars, showing car interiors or overly close-up details, or containing heavily obstructed views. First, we remove duplicates based on the Euclidean distance between image embeddings obtained from a pretrained CLIP model [51], with a threshold of 0.9. Second, we apply the YOLOv11x model [25], pretrained on the COCO dataset [31], to detect car instances. We retain images that include at least one car with a confidence score of 0.6 or higher and a bounding box of at least 64 pixels on each side. Third, we use Qwen2.5-7B [70] to remove images that depict car interiors, overly detailed or abstract representations, or toy models, ensuring that the retained images feature recognizable, real cars. Fourth, we detect and blur any visible faces using DataComp's approach [15] to address privacy concerns. Finally, we eliminate images with strongly overlapping bounding boxes to reduce ambiguity during annotation. We segment cars with SAM 2 [52] and calculate the overlap between each car and neighboring boxes. We use a validation set of 600 images with overlapping detections to determine the optimal removal threshold. The filtered dataset contains 5.87M car crops.

**Data annotation.** We propose a semi-automatic annotation pipeline that combines the zero-shot classification capabilities of vision-language models (VLM) [57], specialized smaller models [62], and a fast manual verification. This pipeline is needed because FGVC remains challenging even for the largest VLMs available [59]. We combine open and focused prompting for labeling. We pass filtered car detections to Qwen2.5-7B [70], which predicts the car model and a confidence score. We match predictions against a predefined list of car models curated from Wikipedia to ensure label consistency. Next, we prompt GPT-4o [1] to confirm the Qwen2.5-7B prediction, or to propose an alternative, and a confidence score. We observed that focused prompting improves recall over open prompting, informing our choice of strategy. Preliminary evaluation of VLM predictions reveals that (1) predictions are often correct, (2) GPT-4o generally outperforms Qwen2.5-7B, with a degree of complementarity between them, and (3) GPT-4o tends to produce overconfident predictions. To further enhance accuracy, we integrate predictions from specialized discriminative models. Specifically, we first pretrain a MoCo v3 model [4] with a ViT-S backbone on all filtered car crops to obtain a car-specific encoder. We then fine-tune the supervised classifiers $DeiT_Q$ and $DeiT_G$ using DeiT [62], with Qwen2.5-7B and GPT-4o predictions respectively as weak labels. To avoid overfitting, we train five instances of each classifier using disjoint 80/20% training/test splits. These models complement VLMs by introducing task-specific discriminators. We manually validate 20 000 instances of 20 car models to determine suitable prediction thresholds for ensembling Qwen2.5-7B, GPT-4o, $DeiT_Q$, and $DeiT_G$. We involve three annotators to perform majority voting. The inter-annotator agreement (IAA), measured with Fleiss' Kappa, is 0.76. We obtain an average accuracy of 98.8% with individual model thresholds of 90%, 90%, 80%, and 80%. As a last step, we manually verify 100 randomly sampled car crops. Again, we involved three annotators, obtained an IAA of 0.79, and performed majority voting. We removed the 14 models with five or more errors (69.1% of the error count) and kept 190 models, achieving a final accuracy of 99.6%.

**The pretraining dataset** comprises images for models not kept in the labeled subset and images collected with queries for car subtypes such as *car, hatchback, sedan, sports car, etc*. We apply the above filtering steps and ensure that the pretraining and labeled subsets do not overlap. We are mainly interested in visual pretraining and do not keep textual metadata. The subset includes 5.1M car crops, 1.9M for the initial 2005-2007 period, and 200K samples for each subsequent year.

**Dataset statistics.** We keep a car model in the labeled subset if it has at least 31 labeled images for at least 5 years. We select 30 examples per year for testing, and use the others for training. The average number of training instances is 244, with a standard deviation of 224. This imbalance contributes to the dataset realism since visual classes have a variable number of instances [33, 75]. The car crops are contributed by 337K unique users, with an average of 17 samples per user. The average user count per model is 1 276, with a standard deviation of 904. These user counts show that CaMiT reflects a socially shared view of the domain.

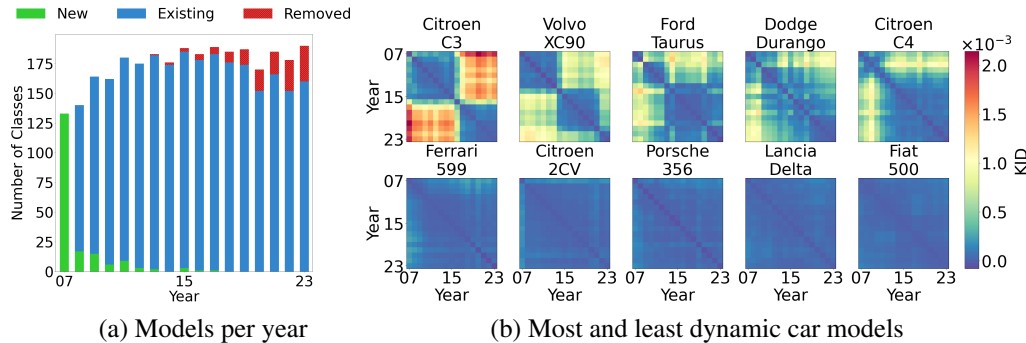

(a) Models per year     (b) Most and least dynamic car models

Figure 3: Dataset analysis across time. We compute CLIP ViT-B embeddings and evaluate the kernel-inception distance (KID) between yearly embeddings.

**Car model temporal dynamics.** We pursue the temporal dynamics analysis summarized in Figure 1 by showing the emergence, evolution, and disappearance of car models from CaMiT in Figure 3 (a). In some cases, rare classes disappear because there are not enough labeled samples, only to reappear in later years. The data stream characteristics from Figure 3 (a) correspond to a generalized continual learning scenario, providing a realistic benchmark for evaluating the influence of time. The embedding analysis in Figure 3 (b) exemplifies some of the most and least dynamic models in CaMiT using the kernel-inception distance (KID) [2], which measures the resemblance between each year-year combination features in the embedding space provided by CLIP ViT-B encoder [51]. KID is more suited than the better-known FID for small samples since it offers an unbiased estimation of the compared sets' resemblance. Changing points generally reflect the appearance of new variants that become popular, such as the *C3 III* for *Citroën C3* in 2016 or the second generation of *Volvo XC90* in 2015. The least dynamic models include classic cars whose design did not change during the period (*Citroën 2CV, Porsche 356*) or changed marginally (*Ferrari 599, Lancia Delta, Fiat 500*). This analysis confirms the prominent influence of car design changes in model depiction shifts over time.

## 4 Experiments

We comprehensively evaluate CaMiT in static pretraining (SPT), time-incremental pretraining (TIP), time-incremental classifier learning (TICL), and time-aware image generation (TAIG) scenarios.

**Evaluation metrics.** We measure classification performance using aggregated accuracy versions. Assuming $i$ and $j$ are the training and test years, we aggregate: $A_{avg}$ - for all $i - j$ year combinations; $A_{crt}$ - for the same $i = j$; $A_{bck}$ - for backward ($i > j$); and $A_{fwd}$ - for forward ($i < j$) in time. When presenting results as matrices, $A_{avg}$ averages all values, $A_{crt}$ aggregates diagonal values, $A_{bck}$ and $A_{fwd}$ average the lower-left and upper-right parts of the matrices. We formally define the accuracy metrics that capture time-related variations in Appendix B. We measure the generation quality using KID [2], with lower values indicating better quality and more diversified synthetic images. We also introduce $A_{avg}^{gen}$, an $A_{avg}$ version using classifiers trained with real images and considering generated images as a test set. Higher $A_{avg}^{gen}$ values indicate better coherence between generated and real images.

**Implementation.** We provide the implementation information and details in different appendix sections. We experiment with ViT-S and ViT-B [10] backbone architectures to ensure comparability

between training strategies and methods. We use self-supervised pretraining, using DINOv2 [46] and CLIP [51], two representative generic models, and MoCo v3 [4] to train a car-specific model. We use CLIP trained with 2B LAION images [58] and the official DINOv2 model trained with 150M images. We chose MoCo v3 for its ease of implementation and good performance. Its initial pretraining uses 1.9M pretraining images from 2005-2007. We train MoCo v3 for 200 epochs, using an AdamW [36] optimizer, a cosine learning rate scheduler with initial $lr = 1.5 \cdot 10^{-4}$ and a batch size of 2048. MoCo v3 yearly fine-tunings include 60 additional epochs with 200K images per year, with an initial $lr = 1 \cdot 10^{-4}$. We implement LoRA [22], noted L, with rank 8 and $\alpha = 16$, updating the attention layers of the ViT architectures in all classification experiments. LoRA yearly updates use the same configuration and training data from the current year ($L_c$) or all preceding and current years ($L_a$). TICL experiments use the methods' original implementations and parameters unless stated otherwise. After pretraining, we run the deterministic methods once. RanPAC [40] and RanDumb [50] are non-deterministic, and we report results averaged over three runs. The two methods are robust, with accuracy variability being under 0.1 points. Note that CLIP with ViT-S is unavailable, and DINOv2 LoRA-adaptation was inconclusive, explaining their absence from the evaluation. In the generation experiment, we use Stable Diffusion 1.5 (SD1.5) [53] and fine-tune it with the CaMiT training subset. It relies on LoRA with rank and $\alpha$ set to 64, trained with a batch size of 64 for 30 epochs, an AdamW optimizer [36] with an initial learning rate of $10^{-4}$, a mixed precision (FP16), a resolution $512 \times 512$ and checkpoints saved every $5\,000$ steps. We used around $1\,300$ A100 GPU hours for all experiments. We detail the experiment costs in Appendix G.

**Statistical significance** tests are performed using the Wilcoxon signed-rank test [69], following common practice in image classification [27, 45, 67]. We apply it to all classifiers compared in Section 4. We report the p-value for the different statistical tests.

### 4.1 Static pretraining effect

Large pretrained models [46, 51] encode generic visual knowledge and provide strong embeddings for numerous classification tasks. A recent result [48] shows that they outperform in-domain training for time-aware recognition of commonsense-level visual classes [54]. A similar comparison does not exist for fine-grained classification, and we perform it using DINOv2 [46], CLIP [51] generic pretraining, and MoCo v3 [4] specific pretraining. We further adapt CLIP and MoCo v3 using LoRA [22] with the 2007 train set ($L_i$). We freeze the pretrained encoders and train a nearest-class mean classifier (NCM) [41] using each year's labeled training data. NCM is a simple yet powerful baseline for image classification with pretrained models, particularly in continual contexts [24, 49].

Table 2: Static pretraining accuracy (in %). #pt - dataset size. $L_i$ - LoRA adaptation with 2007 data. We define accuracy variants in Appendix B. **Best results** in bold, reported separately for ViT-S/B.

| | ViT-S | | | ViT-B | | | | |
|---|---|---|---|---|---|---|---|---|
| | DINOv2 | MoCo v3 | MoCo v3+$L_i$ | DINOv2 | CLIP | CLIP+$L_i$ | MoCo v3 | MoCo v3+$L_i$ |
| **#pt (M)** | 150 | 1.9 | 1.9 | 150 | 2000 | 2000 | 1.9 | 1.9 |
| $A_{avg}$ ↑ | 20.9 | 55.9 | **64.9** | 26.1 | 51.5 | 65.6 | 62.4 | **66.0** |
| $A_{crt}$ ↑ | 25.9 | 65.9 | **75.7** | 32.6 | 61.5 | 74.0 | 73.0 | **76.5** |
| $A_{bck}$ ↑ | 20.9 | 54.0 | **62.6** | 26.1 | 50.9 | **63.9** | 59.7 | 63.2 |
| $A_{fwd}$ ↑ | 20.2 | 56.6 | **65.8** | 25.3 | 50.8 | 66.3 | 63.9 | **67.4** |

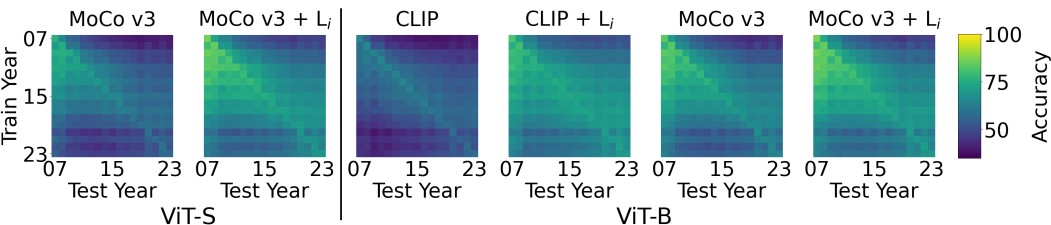

Figure 4: Static pretraining accuracy (in %) for all train-test year combinations.

**Performance degradation occurs when testing with past and future data**. The class representation shift explains this phenomenon in both temporal directions, as demonstrated by lower $A_{bck}$ and $A_{fwd}$ compared with $A_{crt}$ in Table 2. Backward degradation reflects the well-documented forgetting phenomenon observed in CL [14, 26]. Forward degradation occurs mainly due to emerging classes

but also by shifting distributions for existing ones. In Figure 4, the performance gap generally grows with the difference between the test and train years, reflecting an increasing temporal shift. The detailed behaviors differ for generic and specialized models. Performance on the same train-test year gradually decreases for the specialized MoCo v3 model due to an increasing obsolescence of the frozen model, and it is more stable but lower for generic models, calling for mitigation measures.

**Specialized and generic pretraining have similar performance** for car models after LoRA adaptation. Generic models cover many domains and are suboptimal at encoding the fine-grained visual differences between car models by default. They benefit more from the domain adaptation with LoRA, but the effect of this adaptation is statistically significant in all cases ($p < 0.01$). With ViT-B, CLIP roughly doubles DINOv2's performance, a difference explained by a larger training set and a training scheme that uses textual and visual modalities instead of visual only. Our results differ from those reported in [48], where generic pretraining worked much better than in-domain training for commonsense-level visual classes. They support the creation of specific models and the adaptation of generic ones for specialized classification tasks. Appendix C details the training procedures and provides more results.

## 4.2  Time-incremental pretraining

Continual pretraining [7] updates the model when novelty occurs using an approach more efficient than full retraining with all data. We test reservoir- and LoRA-based approaches, and their combination. The reservoir-based update for time-incremental pretraining was proposed in [16]. It mixes a fraction of past data and all samples for each new year, with new samples in the majority. This mix leads to a variable training dataset size, making it difficult to compare the results obtained with model versions. We perform yearly updates with a minority of new samples, a choice that accounts for the accumulation of past data and facilitates comparability across years. We start with the 2005-2007 MoCo v3 ViT-S model pretrained for the SPT experiment, fix the training budget to 1.9M images, and add 200K new samples yearly, sampled from the labeled training and the pretraining subsets. The removed and the new data are selected randomly. We perform LoRA adaptation using each year's labeled training data as an alternative to the reservoir-based approach. Finally, we add a LoRA component to reservoir-based updates, using all labeled data from the reservoir. We train NCM classifiers with the labeled training data corresponding to the pretraining year.

**All TIP variants improve performance over static pretraining,** with statistically significant differences ($p < 0.01$). Interestingly, LoRA-based yearly adaptations in Table 3 work much better than reservoir-based updates. Reservoir and LoRA combination brings further gains. Accuracy increases for backward and forward testing, reducing the negative effects of data shifts compared with SPT. The improvement is larger for past than future data due to knowledge accumulation in the backbones over time. Figure 5 (a) shows that the positive effect lasts when the train-test gap increases.

Table 3: TIP accuracy (in %). $R$: reservoir training. $L_c$ and $L_a$: LoRA adaptation using only the current year's or all known years' labeled training data. **Best results** in bold, reported separately for ViT-S/B .

|  | ViT-S | | | ViT-B | |
|---|---|---|---|---|---|
|  | MoCo v3+$R$ | MoCo v3+$L_c$ | MoCo v3+$R$+$L_a$ | CLIP +$L_c$ | MoCo v3+$L_c$ |
| $A_{avg}$ ↑ | 58.3 | 76.5 | **78.5** | **76.5** | 75.3 |
| $A_{crt}$ ↑ | 69.0 | 88.3 | **90.2** | **87.1** | 86.7 |
| $A_{bck}$ ↑ | 59.9 | 80.4 | **82.5** | **79.7** | 79.2 |
| $A_{fwd}$ ↑ | 55.4 | 71.1 | **73.0** | **71.9** | 70.0 |

Our results are aligned with those in [16], confirming the TIP's usefulness, but at a cost. LoRA-adaptation requires significantly fewer resources than the reservoir-based version and should be privileged. The corresponding yearly updates require 0.3 and 18 GPU hours. While useful, TIP still degrades past and future test data performance, requiring further investigation of temporal data shift mitigation. Appendix D details the TIP training procedures and provides more results.

## 4.3  Time-incremental classifier learning

This approach uses a frozen backbone and updates the classification layer when new data arrives, allowing the classifiers to accumulate knowledge over time. It complements TIP, which updates the model and trains the classifier with new data only due to an evolving encoder. TICL proves particularly effective  [40, 50], and also efficient, motivating the choice of these methods over others from the literature [65]. We combine pretrained models with the following algorithms: NCM,

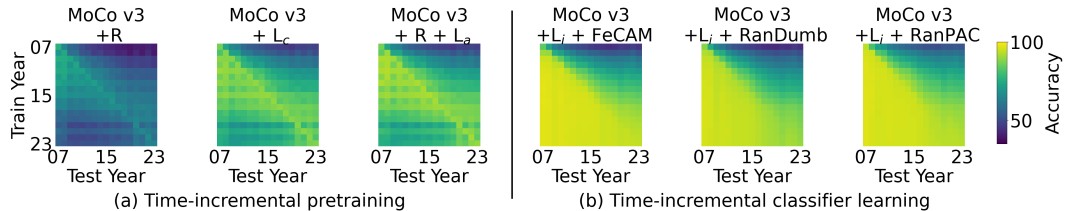

Figure 5: Detailed accuracy (in %) for classification across years using a ViT-S backbone.

FeCAM [18], RanPAC [40], and RanDumb [50]. All algorithms adapt class prototypes incrementally to encode past knowledge, except for NCM. FeCAM, originally proposed for class-incremental learning where class samples arrive simultaneously, stores a class prototype and a covariance matrix per class of size $d^2$, with $d$ the embedding dimensionality. We adapt it to our setting, where class training samples appear gradually. The sizes of past and new samples determine the weights of previous and current years in the prototype and the matrix using a linear weighting. Similarly, we modify NCM in NCM-TI to integrate new samples into existing prototypes based on their proportion. RanPAC [40] and RanDumb [50] use random projections to a high-dimensional space ($r > d$) before prototype-based classification. We experimented with $r = 10\,000$ and $d = 384$ or $d = 768$, depending on the ViT. RanPAC and RanDumb need a projection matrix of size $r^2$. The algorithms' yearly updates require between 3 and 5 minutes.

Table 4: Time-incremental classifier accuracy ($A_{avg}$ in %) with TICL algorithms. $L_i$ - LoRA adaptation with 2007 data. **Best results** are in bold for each pretrained backbone.

|  | ViT-S | | | ViT-B | | | | |
|---|---|---|---|---|---|---|---|---|
|  | DINOv2 | MoCo v3 | MoCo v3+$L_i$ | DINOv2 | CLIP | CLIP+$L_i$ | MoCo v3 | MoCo v3+$L_i$ |
| **NCM** | 20.9 | 55.9 | 64.9 | 26.1 | 51.5 | 65.6 | 62.4 | 66.0 |
| **NCM-TI** | 26.3 | 64.4 | 71.4 | 31.9 | 57.8 | 70.1 | 69.6 | 72.1 |
| **FeCAM** | 61.0 | 78.5 | 85.6 | 64.9 | 68.8 | 79.9 | 81.1 | 81.5 |
| **RanDumb** | 62.1 | **82.2** | 83.1 | 65.4 | 71.6 | 77.2 | **83.4** | 84.2 |
| **RanPAC** | **66.4** | 72.7 | **86.6** | **70.8** | **76.3** | **80.3** | 81.4 | **87.8** |

**TICL provides the best accuracy across time** among all time-related adaptations of backbones and training strategies tested. The $A_{avg}$ gain in Table 4 over the best SPT results from Table 2 exceeds 20 points for both ViT architectures. The improvements over the best TIP results (Table 3) reach 8.1 and 11.3 points for ViT-S and ViT-B, respectively. These gains are statistically significant in all cases ($p < 0.01$). When comparing pretraining strategies, we find that the LoRA-adapted MoCo v3 is best, followed by LoRA-adapted CLIP (for ViT-B). This result confirms the interest of specialized pretraining and parameter-efficient domain adaptation. TICL algorithms are particularly useful for DINOv2 encoders, whose accuracy roughly triples. Method-wise, RanPAC is best, except for the non-adapted MoCo v3, where RanDumb is better. Even NCM-TI, a simple NCM adaptation, provides significant gains compared with NCM. The reported gains come with significant overhead, with FeCAM, RanPAC, and RanDumb requiring over 100M additional parameters.

**TICL methods ensure positive backward and forward transfer [35],** as shown by improved performance in all regions of the matrices from Figures 4 compared with Figure 5. The progressive classifier consolidation explains this behavior by integrating new samples over time. The improvements are particularly large backwards, with past scores often surpassing the current accuracy. These gains occur because some current samples depict older model variants, reinforcing the classifier's representation of the past. This finding was not previously reported in the CL literature, highlighting the importance of experimenting with realistic time-aware scenarios. We also observe positive forward transfer, with smaller gains when the future gap increases. We expect such behavior for time-dependent concepts due to new models or designs appearing periodically. Our findings confirm the need for encoder and/or classifier updates to keep pace with the concept dynamics in fine-grained classification. Appendix E details the method adaptation and training, and provides further results.

### 4.4 Time-aware image generation

Standard pipelines use the relation between image pixels and captions to generate images [53], and do not explicitly model the temporal aspect. We introduce TAIG to test whether such modeling is

beneficial when generating content for visual concepts whose appearance shifts over time. We use the CaMiT training subset to fine-tune SD1.5 [53] with LoRA [13], using the following time-aware captions: ``A photo of CAR_MODEL in YEAR'', where CAR_MODEL is the model name and YEAR is the photo publication year. We also create caption variants without the year to better discern the temporal effects. We append $TAIG$ and $FT$ to the original diffusion model name to name the fine-tuned models obtained with and without year mentions in the captions. In Table 5, we evaluate temporal effects by prompting the original and fine-tuned models with the same captions used during TAIG fine-tuning to generate 30 images per model-year combination.

**Time-awareness improves image generation** as shown by the lower KID and higher accuracy obtained by SD1.5$_{TAIG}$. The improvements over SD1.5 are statistically significant in both cases ($p < 0.01$). The year-agnostic fine-tuning improves over standard generation, but the gains are smaller than for SD1.5$_{TAIG}$. The gains occur even though SD1.5$_{FT}$ has more samples per class since it aggregates the training images across all years.

Table 5: KID and $A_{avg}^{gen}$ (in %) scores for standard generation (SD1.5) and fine-tuning without or with temporal metadata (resp. SD1.5$_{FT}$ and SD1.5$_{TAIG}$).

| Model | KID ($\times 10^{-4}$) ↓ | $\mathbf{A}_{avg}^{gen}$ ↑ |
|---|---|---|
| SD1.5 | 6.83 | 46.1 |
| SD1.5$_{FT}$ | 4.48 | 52.1 |
| SD1.5$_{TAIG}$ | **4.19** | **54.1** |

The results indicate that considering temporal metadata during generator training is beneficial, even using a simple fine-tuning. We hope this finding will encourage more refined attempts to include temporal awareness in content generation pipelines. We detail the fine-tuning training in Appendix F.

## 5   Limitations

CaMiT represents car models in the long term, offering a challenging playground for time-aware fine-grained image classification and generation. However, we acknowledge limitations related to the dataset itself and the experiments. As with any dataset, CaMiT is affected by a selection bias [12] that manifests at several levels. First, we used Flickr to collect images, raising a generalization question. We made this choice after inconclusively trying other data sources. Notably, Google Images' temporal metadata was imprecise, and TIC-DataComp has sufficient images only from 2017 onward. Flickr is a socially relevant choice since it gathers contributions from a large user sample, with 337K users contributing to CaMiT. Second, the dataset covers 17 and 19 years, for its labeled and unlabeled parts, roughly three times longer than TIC-DataComp. Including a more distant past is desirable but difficult in practice since the amount of digitized photos is insufficient to represent the car models. Third, the class selection depends on the number of Flickr images available per model. Some car models have more samples than others among the 190 classes, and the training set is imbalanced. This imbalance induces performance challenges but augments the realism of the tested classification and generation tasks. Finally, the geographic spread of brands and models is imbalanced, with Europe dominating the dataset (27 out of 48 brands, 99 out of 199 models). This imbalance comes from the car production per region during the period, but also reflects Flickr's geographic biases [61].

We do not have the resources to annotate the entire dataset and resort to semi-automatic labeling. We acknowledge that the semi-automatic pipeline entails biases, which could potentially affect the dataset distribution. However, we carefully design multiple filters and use several classifiers to boost the accuracy of the annotation process. The verification done with a subset shows that the annotations are 99.6% correct, confirming the labeling reliability.

The temporal annotations in CaMiT may capture both genuine design evolution (new model variants) and physical aging (older cars photographed years after their release). This conflation can blur temporal modeling results. Future work could disentangle these two factors by aligning timestamps with official model release dates or registration metadata, allowing more precise study of design changes over time.

We distribute image links, embeddings, and metadata, but not the images themselves, to comply with copyright regulations as in [15, 58]. We initially tried to include only images with redistributable licenses, but they were insufficient. We also distribute image embeddings to improve long-term usability. Some image links will become inactive over time, requiring future work to rerun experiments for comparability. We measure link persistence by recollecting 10K images six months after the initial process. Missing links amount to only 0.68%, hinting at long-term CaMiT usability.

Image generation entails potential misuse, for instance, to enable misidentification in insurance fraud systems. Applying the method to more sensitive images, such as human faces, could have a significant negative societal impact but would require a considerable data collection effort. However, the fine-tuned and standard diffusion models share these risks.

We illustrate different dataset usage scenarios with sound existing methods and their adaptations. We acknowledge that more refined modeling is possible, but it is beyond the immediate scope. We encourage other teams interested in the topic to delve deeper into the proposed scenarios' methodological aspects and introduce new ones based on CaMiT's rich content.

# 6    Conclusion

We introduced CaMiT, a new time-aware dataset designed for fine-grained image classification and generation. It includes 190 car models photographed over 17 years and a pretraining dataset covering 19 years, contributed by 337K unique users. We first analyzed the temporal car dynamics and showed that a temporal data shift occurs at a pace depending on the model. This shift calls for time modeling in AI classification and generation models to reflect the changing class appearance. We then presented comprehensive classification experiments with SPT, TIP, and TICL. SPT degrades performance backward and forward in time while TIP and TICL mitigate this effect. TICL is more effective, particularly backwards, and offers an interesting effectiveness-efficiency balance. We show that pretraining is competitive in fine-grained image classification, a result at odds with previous findings for commonsense-level classification [48]. Finally, we introduce time-aware image generation, highlighting the positive effect of considering temporal metadata when training a generator for changing visual classes. We hope that CaMiT availability stimulates the AI community's interest in integrating time in visual classification and generation models to avoid their obsolescence.

# 7    Acknowledgement

This work was financially supported by the Agence de l'Innovation de Défense (AID) and partly supported by the SHARP ANR project ANR-23-PEIA-0008 in the context of the France 2030 program. Funded by the European Union. Views and opinions expressed are however those of the authors only and do not necessarily reflect those of the European Union nor the European Commission. Neither the European Union nor the granting authority can be held responsible for them. This work was supported under the EDF Project FaRADAI (grant number 101103386). This work was made possible by the FactoryIA supercomputer (funded by the Ile-de-France Regional Council) and by HPC ressources from GENCI-IDRIS(Grant 2024-AD011015944).

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
