# Appendix for "CaMiT: A Time-Aware Car Model Dataset for Classification and Generation"

Frédéric Lin    Biruk Abere Ambaw    Adrian Popescu    Hejer Ammar
Romaric Audigier        Hervé Le Borgne

Université Paris-Saclay, CEA, List, F-91120, Palaiseau, France
{firstname.lastname}@cea.fr

# 1 Appendix Contents

Submitted to 39th Conference on Neural Information Processing Systems (NeurIPS 2025). Do not distribute.

# A Dataset creation

## A.1 A1: Data collection

### A.1.1 Google Images

We initially experimented with Google Images as a data source, leveraging its extensive web index and the ability to filter results by date using Google Custom Search Engine (CSE) APIs [8]. Our goal was to assess whether Google could provide reliable, fine-grained car model images with usable temporal metadata.

We implemented a search script using the `googleapiclient` package [7] provided by the Google Custom Search service. For each car model (e.g., "Renault Twingo"), we divided each year into 5 time intervals (roughly bimonthly) and queried with a temporal constraint in the form `before:{end_date} after:{start_date}`.

To assess the reliability of the temporal metadata associated with Google Image results, we implemented a verification pipeline using the `contextLink` field returned by the Google Custom Search API. For each image, we extracted the URL of the host page and attempted to retrieve an explicit publication date using multiple strategies.

We parsed each web page using `BeautifulSoup` [22], looking for common metadata fields such as:

- `<meta name="date">`, `<meta property="article:published_time">`, `<meta itemprop="datePublished">`
- JSON-LD blocks with `"datePublished"` keys
- `<time>` or `` tags with `datetime`

If structured metadata is unavailable, we applied pattern-matching heuristics using regular expressions to extract date-like substrings from the page content and URL itself.

We performed this analysis using a representative query: "Renault Twingo". After collecting and validating publication dates from the image context links for this model, we compared them to the intended query year. For each image, we computed the difference between the target year and the actual year extracted from metadata.

We then conducted two statistical tests to assess whether this difference was significantly different from zero: (1) a one-sample t-test and (2) a Wilcoxon signed-rank test [26] . Both tests revealed strong evidence of temporal mismatch. The t-test returned $p = 1.58 \times 10^{-47}$, and the Wilcoxon test returned $p = 5.41 \times 10^{-70}$. These results indicate that the actual publication years significantly differ from the intended targets.

We also found that 49.55% of images retrieved for "Renault Twingo" lacked any reliably extractable publication date, either due to absent metadata or ambiguous timestamps. Figure 1 shows the distribution of year differences for the subset with valid timestamps. The histogram confirms a substantial deviation from the target year, with many images appearing earlier or later than intended.

While these results are limited to a single model, they highlight the unreliability of temporal filtering in the Google Custom Search API and motivated our decision to switch to a more structured and timestamp-consistent source.

### A.1.2 TIC-DataComp

We also considered TIC-DataComp [6], the time-aware version of DataComp [5], as a potential resource for collecting car model images. DataComp includes metadata for 12.8 billion images uploaded to the Web between 2014 and 2023. We matched car model names against the metadata and retrieved over 90 million potentially relevant samples, averaging 475K samples per car model with a standard deviation of 503K.

However, the temporal distribution of images in TIC-DataComp is highly imbalanced (see Figure 2), with the majority of samples concentrated between 2017 and 2022. For instance, the dataset contains only 256M images for 2014 and just 193K for 2023, compared to over 2 billion for years like 2019 and 2021. This severe imbalance, combined with the relatively narrow coverage (10 years), limits its

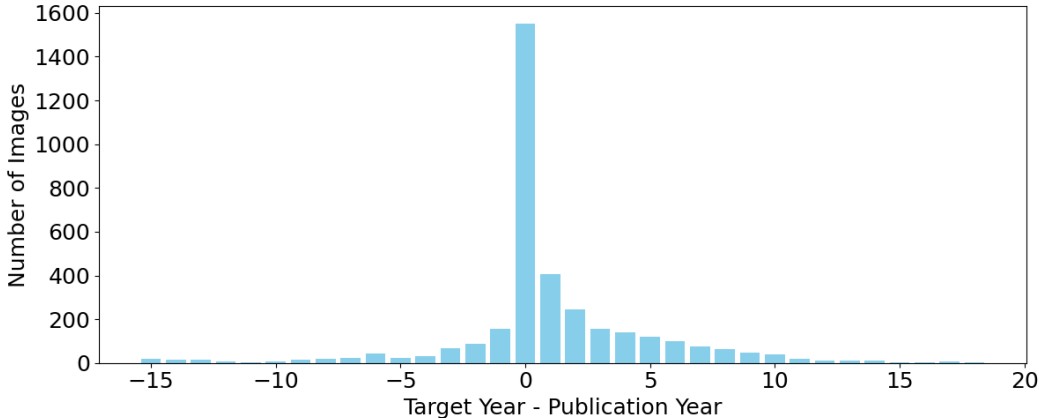

Figure 1: Histogram of differences between the target query year and the extracted publication year for "Renault Twingo" images with valid metadata.

usefulness for studying temporal dynamics in car model imagery. In contrast, our labeled dataset spans 17 years (19 years for the unlabeled dataset), enabling a broader and more consistent temporal analysis.

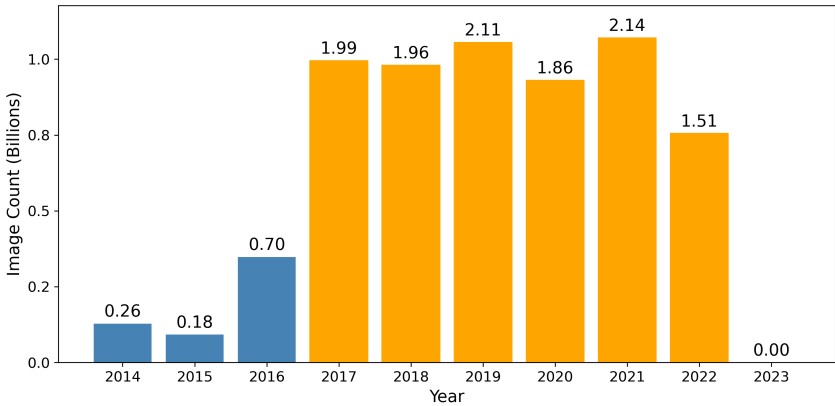

Figure 2: Yearly distribution of images in TIC-DataComp. Most data is concentrated between 2017 and 2022, with earlier and later years significantly underrepresented.

### A.1.3 Flickr

Flickr is a multimedia content sharing platform that has been active since 2004. It has been extensively used to curate visual datasets [4, 11, 13, 14, 19, 24, 27]. Flickr provides access to its public photos via an API, combining text and different types of metadata, including temporal ones. We queried the API using the `flickr.photos.search` method with the car-related classes as `text`, temporal intervals delimited with `min_taken_date` and `max_taken_date`, and the `relevance` filter to sort the results. We collected the following metadata: `id`, `owner`, `title`, `tags`, `date_taken`, `date_upload`, `license`, and `url_z`. After `id`-based deduplication, we proceeded to collect up to 5000 images per query-year combination. Our objective was to obtain a large domain coverage, and we built 425 unique queries using Wikipedia, ImageNet, and by prompting ChatGPT. The list comprises car subtypes, brands, and models. A few examples of subtypes include: `hatchback`, `limousine`, `pickup`, `sedan`, `subcompact`, `supercar`, and `SUV`. Examples of brands include: `Toyota`, `Volkswagen`, `Daihatsu`, `Buick`, `Bugatti`, `Dacia`, and `Xpeng`. We list the 190 car models included in the labeled CaMiT subset in Table 1.

Table 1: List of car models in CaMiT grouped by brand.

| Brand | Models |
|-------|--------|
| Acura | acura mdx, acura tlx |
| Alfa Romeo | alfa romeo giulietta, alfa romeo mito |
| Aston Martin | aston martin db9, aston martin dbs, aston martin rapide, aston martin vantage |
| Audi | audi a3, audi a4, audi a5, audi a6, audi a8, audi q3, audi q5, audi q7, audi r8, audi tt |
| Bentley | bentley bentayga, bentley continental |
| Bmw | bmw 1 series, bmw 3 series, bmw 5 series, bmw 7 series, bmw x1, bmw x3, bmw x5, bmw x6, bmw z4 |
| Bugatti | bugatti chiron, bugatti veyron |
| Cadillac | cadillac cts, cadillac escalade |
| Chevrolet | chevrolet camaro, chevrolet corvette, chevrolet impala, chevrolet suburban |
| Chrysler | chrysler 300 |
| Citroën | citroen 2cv, citroen c3, citroen c4 |
| Dodge | dodge challenger, dodge charger, dodge durango, dodge viper |
| Ferrari | ferrari 458 italia, ferrari 599, ferrari f40, ferrari f430 |
| Fiat | fiat 500, fiat panda, fiat punto |
| Ford | ford bronco, ford crown victoria, ford ecosport, ford edge, ford escape, ford explorer, ford fiesta, ford focus, ford fseries, ford gt, ford mondeo, ford mustang, ford ranger, ford taurus |
| Honda | honda accord, honda civic, honda crv, honda fit, honda integra, honda nsx, honda odyssey, honda s2000 |
| Hyundai | hyundai accent, hyundai elantra, hyundai genesis coupe, hyundai santa fe, hyundai sonata, hyundai veloster |
| Infiniti | infiniti g |
| Isuzu | isuzu dmax |
| Jaguar | jaguar etype |
| Jeep | jeep cherokee, jeep grand cherokee, jeep wrangler |
| Kia | kia optima, kia rio, kia sorento, kia soul, kia sportage, kia stinger |
| Lamborghini | lamborghini aventador, lamborghini gallardo, lamborghini huracan, lamborghini murcielago |
| Lancia | lancia delta |
| Land | land rover defender, land rover discovery, land rover range rover |
| Lexus | lexus gs, lexus is, lexus rx |
| Lincoln | lincoln continental |
| Maserati | maserati granturismo, maserati quattroporte |
| Mazda | mazda cx5, mazda mazda3, mazda mazda6, mazda mx5, mazda rx7 |
| Mercedes-Benz | mercedes-benz cclass, mercede-sbenz cla, mercedes-benz cls class, mercedes-benz eclass, mercedes-benz gclass, mercedes-benz sclass, mercedes-benz slr mclaren |
| Mini | mini countryman, mini cooper |
| Mitsubishi | mitsubishi eclipse, mitsubishi lancer evolution, mitsubishi outlander, mitsubishi pajero |
| Nissan | nissan 350z, nissan 370z, nissan altima, nissan gtr, nissan maxima, nissan pathfinder, nissan qashqai, nissan skyline gtr |
| Opel | opel astra, opel corsa, opel insignia, opel zafira |
| Pagani | pagani huayra, pagani zonda |
| Peugeot | peugeot 207, peugeot 208 |
| Porsche | porsche 356, porsche 911, porsche cayenne, porsche cayman, porsche macan, porsche panamera |
| Renault | renault clio, renault kangoo, renault megane, renault scenic |
| Rolls-Royce | rolls-royce phantom |
| Seat | seat ibiza, seat leon |
| Skoda | skoda fabia, skoda octavia, skoda superb |
| Smart | smart fortwo |
| Subaru | subaru brz, subaru forester, subaru impreza, subaru legacy, subaru outback |
| Suzuki | suzuki swift |
| Tesla | tesla model s, tesla roadster |
| Toyota | toyota 4runner, toyota camry, toyota land cruiser, toyota prius, toyota rav4, toyota sienna, toyota supra, toyota tacoma, toyota tundra, toyota yaris |
| Volkswagen | volkswagen beetle, volkswagen golf, volkswagen passat, volkswagen polo, volkswagen scirocco, volkswagen tiguan, volkswagen touareg, volkswagen transporter |
| Volvo | volvo s40, volvo s60, volvo s80, volvo v40, volvo v60, volvo xc60, volvo xc90 |

## A.2   Data Filtering

Our filtering pipeline removes duplicate, irrelevant, and low-quality images to prepare a dataset suitable for fine-grained car model recognition. This section expands upon the overview presented in the main text (Section 3.2), providing a detailed account of each step in the filtering process. We began with 7.5M images collected via the Flickr API. They were processed through multiple stages of filtering, as summarized in Table 2.

First, we removed duplicate images based on the Euclidean distance between image embeddings extracted from a pretrained CLIP model [20], using a threshold of 0.9. Next, we applied the YOLOv11x model [12], pretrained on the COCO dataset [14], to detect car instances. Only images containing at least one valid car detection were retained, resulting in 4.9M unique images and a total of 13.22M car bounding boxes. To ensure detection quality, we applied a confidence threshold of 0.6 and discarded bounding boxes with width or height below 64 pixels, reducing the dataset to 6.97M bounding boxes.

Next, we addressed the semantic quality of the detections. While YOLO provided spatial bounding boxes for cars, some detections corresponded to irrelevant content such as vehicle interiors, close-up parts, toy models, or artificial renderings. To filter these cases, we used the Qwen2.5-7B vision-language model [2], guided by the following structured prompt:

```
Please analyze the image and answer the following three
questions:
1. Does the image only show the interior of the car with
little to no exterior visible?
- Answer with true if it is an interior view, otherwise false.
2. Is the image too zoomed-in, showing only a small portion
of the car, making it difficult to recognize its model?
- If the image only contains a tiny part of the car (e.g.,
just a wheel, headlight, or badge) and makes model recognition
difficult, answer true.
- If the car is not fully visible but still recognizable,
answer false.
3. Does the image show a toy car, scale model, or artificial
rendering rather than a real vehicle?
- Answer true if the car is a miniature or synthetic
representation; otherwise, false.
Return your answers in the following JSON format:
{"interior": <true/false>, "zoomed_in": <true/false>, "toy":
<true/false>}
```

We discarded any image for which at least one of the three fields was marked as `true`. This step allowed automatically removing interior views, extreme close-ups, and non-realistic content. After this semantic filtering stage, 6.37M car bounding boxes remained.

We applied face detection and blurring following the method described in DataComp. While this did not alter dataset size, it ensured adherence to privacy standards.

As a final step, we introduced a directional occlusion filtering mechanism to reduce annotation ambiguity caused by overlapping car detections. This step targets car detections where one car significantly occludes another, which may hinder accurate labeling. To quantify occlusion, we defined a containment ratio $C$ between a given bounding box $b_1$ and any other car segmentation mask $m_2$ (generated using Segment Anything 2 [21]) within the same image:

$$C = \frac{|m_2 \cap b_1|}{|b_1|} \tag{1}$$

where $|\cdot|$ denotes the area in pixels.

Equation 1 captures the extent to which $b_1$ is obscured by $m_2$, offering a directional alternative to standard IoU. Unlike IoU, it explicitly models asymmetric occlusion—particularly useful in crowded scenes where smaller or background cars may be partially hidden by larger foreground vehicles.

To calibrate a suitable threshold for filtering, we manually annotated 600 image crops sampled evenly across six containment bins (0.0 to 0.5+, 100 samples per bin). Each crop was labeled as either "highly occluded" or "not occluded." Figure 3 summarizes the number of high-occlusion cases per bin. Based on this analysis, we selected a threshold of 0.2: occlusion cases become significantly more frequent above this value (e.g., 28 or more per bin), while only 3 of the 200 samples below it were annotated as highly occluded—yielding a false positive rate of just 0.5%.

Applying this final filter removed approximately 497K ambiguous crops, resulting in 5.87M high-quality car instances.

Table 2: Summary of dataset size after each filtering step.

| Filtering Step | Remaining Instances |
|---|---|
| Raw images collected | 7.5Mimages |
| After deduplication and car detection | 4.9M images |
| Initial car bounding boxes | 13.22M boxes |
| After score/size thresholding | 6.97M boxes |
| After Qwen2.5-7B semantic filtering | 6.37M boxes |
| After overlap removal (SAM 2) | 5.87M boxes |

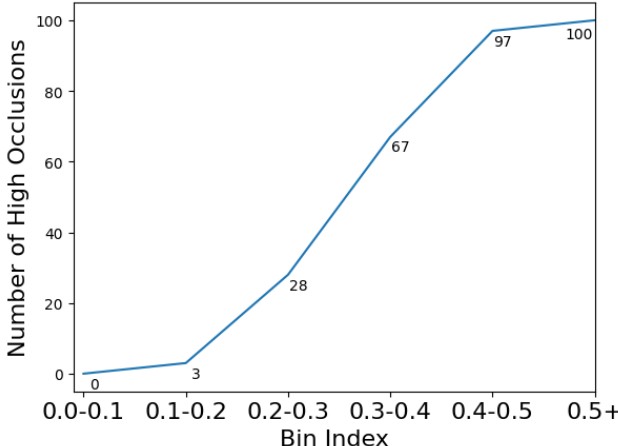

Figure 3: Number of highly occluded crops across containment ratio bins. A sharp increase in occlusion cases occurs above 0.2, motivating its selection as the filtering threshold.

## A.3 Data Annotation

To obtain high-quality fine-grained vehicle classification labels, we design a semi-automatic annotation pipeline leveraging both vision-language models (VLMs) and specialized discriminative models.

**Step 1: Zero-shot car model labeling with Qwen2.5-7B.** Each car crop is passed to Qwen2.5-7B using a structured prompt:

```
Please identify the brand and model of the car in the image. If the
car is unrecognizable, respond with 'unknown'. The response should
be in JSON format as follows: {'brand': '...', 'model': '...',
'confidence': '...'}
```

This stage yields 3.49M non-unknown bounding boxes.

**Step 2: Filtering to ensure representative distributions.** To ensure each class is both sufficiently represented and reliably annotated, we filter out brand-model pairs having less than 2,000 total crops

and discard year-specific subsets with less than 40 samples. This step serves two purposes: it avoids training on underrepresented classes likely to consist of outliers or rare edge cases, and it reduces the risk of including erroneous labels from Qwen2.5-7B, which are more frequent in sparsely occurring categories.

In addition, to improve consistency and reduce spurious predictions, we match Qwen predictions against a curated list of car models derived from Wikipedia. Starting from the full set of approximately 45,000 unique brand-model combinations produced by Qwen, we apply string parsing and normalization techniques (e.g., case folding, removal of extraneous tokens) to align predictions with canonical class names. This filtering step reduces the space of valid class labels to around 300 representative car models. After these filtering stages, we retain 1.88M Qwen-annotated boxes with improved label quality and distributional reliability.

**Step 3: Car Model Confirmation with GPT-4o.**  For each Qwen-labeled crop, we prompt GPT-4o [1] using the following setup:

- **System prompt:**

```
Analyze the image and return:  {
  "model":  string, // Identified car model (return provided class
if it matches)
  "model_probability":  number, // Probability (0-100) of the
predicted model
  "car_probability":  number // Probability (0-100) that the object
is a real car
}
Strict JSON format required.
```

- **User prompt:** The car model predicted by Qwen2.5-7B.

This stage yields 1.87M non-`unknown` predictions from GPT-4o. During early experimentation, we performed qualitative comparisons between Qwen2.5-7B and GPT-4o on a range of representative samples. Visual inspection suggested that GPT-4o generally produced more accurate predictions and complemented Qwen2.5-7B, particularly on difficult examples. We also observed that prompting GPT-4o with Qwen's prediction (i.e., focused prompting) led to more reliable outputs than open-ended prompts.

**Step 4: Agreement filtering.**  To increase label precision, we apply two filtering stages based on the outputs of Qwen2.5-7B and GPT-4o:

- **Intersection:** retain only the 1.36M boxes for which both models produce valid, non-`unknown` predictions.
- **Strict agreement:** from the intersection set, retain only the 963K boxes where both models predict the exact same brand and model.

**Step 5: Representation learning with MoCo v3.**  To build a car-specific visual encoder, we pretrain a MoCo v3 [3] model with a ViT-S backbone on all 5.87 million filtered car crops, including unlabeled boxes. The model is trained for 300 epochs using the AdamW [15] optimizer with a learning rate of 1.5e-4 and a weight decay of 0.1. We adopt a cosine annealing learning rate schedule with 40 warmup epochs, and use a batch size of 2048 to ensure stable training at scale.

**Step 6: Fine-tuning with automatic labels.**  We fine-tune the pretrained encoder using two variants of the DeiT [25] classifier: $DeiT_Q$, trained on Qwen-labeled data, and $DeiT_G$, trained on GPT-4o-labeled data. This stage leverages the 1,356,731 boxes for which at least one model provided a confident label, without requiring strict agreement between Qwen and GPT-4o. To improve robustness and mitigate overfitting, we perform five independent training runs with disjoint 80/20 train/test splits. Each classifier is fine-tuned for 150 epochs using the AdamW optimizer with a learning rate of 5e-4, a cosine annealing schedule with 5 warmup epochs, and a batch size of 512.

*Note:* These models are used solely for annotation and data cleaning. They are not used in any of the experimental evaluations (e.g., SPT, TIP, TICL) to ensure a fair and unbiased assessment of the methods under study.

**Step 7: Consensus filtering with discriminative models**   After fine-tuning the discriminative classifiers, we apply all four models—Qwen2.5-7B, GPT-4o, DeiT$_Q$, and DeiT$_G$—to the annotated dataset and retain only the bounding boxes where all models agree on the predicted brand and model. This consensus step further increases label reliability by ensuring agreement across both generalist and specialized systems. As a result, we obtain 849K high-confidence annotations for 204 classes.

**Step 8: Manual validation and final refinement**   To evaluate the quality of the consensus-labeled dataset, we randomly sample 20,000 bounding boxes across 20 classes, specifically selecting only high-confidence predictions. Confidence scores for Qwen2.5-7B and GPT-4o are obtained directly from their structured prompts, where each model outputs an explicit confidence estimate along with the predicted label. For the discriminative classifiers (DeiT$_Q$ and DeiT$_G$), we compute confidence as the softmax probability associated with the predicted class (i.e., the maximum logit after softmax). We select samples where Qwen $\geq$ 90%, GPT-4o $\geq$ 90%, DeiT$_Q$ $\geq$ 85%, and DeiT$_G$ $\geq$ 85%.

Manual inspection is conducted using a custom annotation interface designed for efficient verification. The interface, shown in Figure 4, displays car crops along with predictions from Qwen2.5-7B and GPT-4o, their confidence scores, and allows annotators to flag errors. Annotators select images for review by clicking and pressing Enter, which visually highlights the selected bounding box. Dropdowns enable rapid switching between car classes and model years. A progress bar at the top provides an overview of annotation coverage.

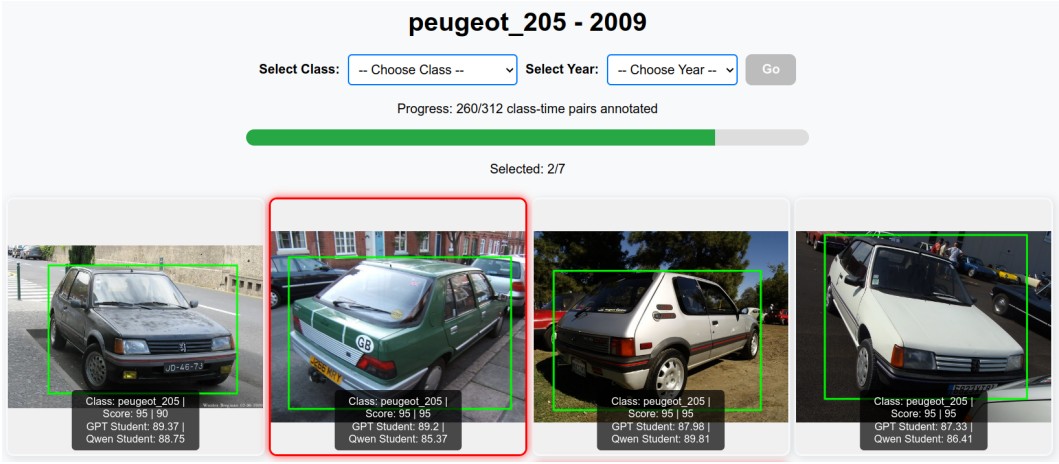

Figure 4: Annotation Interface for Fine-Grained Vehicle Classification. The interface displays car images with bounding boxes, predictions from Qwen2.5-7B and GPT-4o, and confidence scores. Users select images for manual verification by clicking on them and pressing Enter, which highlights the selected image with a red border. Dropdown menus at the top allow navigation between different car classes and years. Progress tracking is shown at the top, indicating the number of annotated class-time pairs.

Manual inspection of the 20,000 samples is performed by three annotators using majority voting, and reveals an average labeling accuracy of 98.8%. Inter-annotator agreement, measured using Fleiss' Kappa, is 0.76. To further assess class-level reliability, we manually annotate 100 images per class across all 204 remaining classes. Again, three annotators perform majority voting, and the inter-annotator agreement for this task is 0.79. Based on this validation, we identify 14 classes with consistent prediction errors, which are subsequently removed from the dataset. This final refinement results in a dataset with an estimated annotation accuracy of 99.6%.

**Step 9: Temporal split with instance-level de-duplication.**   To construct time-aware train/test splits for evaluation, we uniformly sample bounding boxes across time while ensuring no image overlap between sets. Specifically, for each model-year pair, we select 30 bounding boxes for the test set. The remaining bounding boxes for the model-year pair are used for training, but we enforce that no training box originates from the same source image as any test box to prevent feature leakage through shared backgrounds or viewpoints. This de-duplication ensures that test-time evaluation

reflects true generalization to unseen instances rather than memorization of shared image content. After applying this sampling strategy across all classes and years, we obtain a total of 693K bounding boxes for training and 94K for testing. These subsets serve as the basis for all time-incremental experiments in the submission.

Throughout all classification experiments, we rely exclusively on cropped bounding boxes, treating each crop as an individual instance. This design choice ensures that models focus on the vehicle itself, reducing background noise and emphasizing class-discriminative features. In contrast, for the time-aware image generation experiments, we use the full original images containing the annotated bounding boxes. This enables modeling of broader visual context—including background scenes, co-occurring vehicles, and environmental cues—which are critical for generating realistic, temporally consistent imagery but less relevant for fine-grained recognition tasks.

## B  Evaluation metrics

We define accuracy metrics to measure the different training-test years combinations occurring in time-aware image classification. We use the following notations: $i$ - current training year, $j$ - current test year, $y_{max} = 2023$ - last training/test year, $y_{min} = 2007$ - first training year $r = y_{max} - y_{min} + 1 = 17$ - total number of training and test years, $A(i, j)$ - the accuracy of a classifier trained on year $i$ and tested on year $j$. Note that we could also refer to an *index* of the year with $y_{min} = 1$ and $y_{max} = 17$, without changing the following.

$$\mathbf{A_{avg}} = \frac{1}{r^2} \sum_{i=y_{min}}^{y_{max}} \sum_{j=y_{min}}^{y_{max}} A(i, j) \tag{2}$$

$$\mathbf{A_{crt}} = \frac{1}{r} \sum_{i=y_{min}}^{y_{max}} A(i, i) \tag{3}$$

$$\mathbf{A_{bck}} = \frac{2}{r(r-1)} \sum_{i=y_{min}+1}^{y_{max}} \sum_{j=y_{min}}^{i-1} A(i, j) \tag{4}$$

$$\mathbf{A_{fwd}} = \frac{2}{r(r-1)} \sum_{i=y_{min}}^{y_{max}-1} \sum_{j=i+1}^{y_{max}} A(i, j) \tag{5}$$

$\mathbf{A_{avg}}$ (Equation 2) aggregates all model training-test year combinations evaluated in the main text. It gives a global view of each tested system. $\mathbf{A_{crt}}$ (Equation 3) focuses on the cases when we train and test with images published the same year ($i = j$). It aggregates scores presented on the matrix diagonals such as those in Figure 5. $\mathbf{A_{bck}}$ (Equation 4) averages the accuracies obtained when $j < i$, aggregating the scores from the bottom-left of the detailed matrices. It provides insights about how well the different tested systems cope with past data. $\mathbf{A_{fwd}}$ (Equation 5) averages the accuracies obtained when $j > i$, aggregating the top-right of the detailed matrices. It reflects the systems' behavior when encountering future test data.

## C  Static Pretraining

### C.1  Static Model Training Procedure

To investigate how static pretrained models handle temporal shifts in fine-grained classification (Section 4.1), we describe our training setup. In this setting, models are pretrained once and evaluated across all years using a frozen encoder and a simple nearest class mean (NCM) classifier [17]. We report accuracy across multiple temporal splits and restrict adaptation to the first year of the labeled set.

**Pretrained encoders.**   We evaluate both generic and domain-specific pretrained encoders:

- **DINOv2 ViT-S/ViT-B:** Official self-supervised models trained on 150M curated images [18].
- **CLIP ViT-B:** Public model trained on 2B LAION images with paired text supervision [20].
- **MoCo v3 ViT-S/ViT-B:** Trained from scratch on CaMiT data (see below).

**MoCo v3 pretraining.**   We follow the original MoCo v3 setup closely to train models from scratch on CaMiT, using 1.9M car instances from years 2005–2007 for 200 epochs. We initially attempted to fine-tune from the ImageNet-1K pretrained checkpoints provided in the official MoCo v3 repository, but observed consistently lower performance compared to training from scratch. As a result, all reported MoCo v3 models are trained from scratch on CaMiT. Our implementation preserves key design choices from MoCo v3, including optimization strategy, augmentations, and learning rate scheduling. The setup includes:

- **Optimizer:** AdamW
- **Batch size:** 2048
- **Learning rate:** $1.5 \times 10^{-4}$
- **Scheduler:** Cosine learning rate decay
- **Augmentations:** A MoCo v3-style augmentation pipeline, including random resized crops, color jittering, Gaussian blur, horizontal flipping, and grayscale conversion.

**Initial LoRA adaptation ($L_i$).**   To enable domain adaptation while preserving encoder generality, we apply LoRA [10] to the pretrained encoders using only the labeled 2007 CaMiT training set. The backbone weights remain frozen, and only the LoRA parameters applied to the self-attention layers are updated. We implement LoRA using the Hugging Face `peft` library [16] , and append a trainable linear classification head to the adapted encoder. Training is supervised with a cross-entropy loss over the car model classes.

The adaptation setup is as follows:

- **Optimizer:** SGD with momentum 0.9
- **LoRA rank:** 8
- **Scaling factor:** 16
- **Dropout:** 0.1
- **Epochs:** 20
- **Batch size:** 48
- **Learning rate:** 0.01
- **Weight decay:** $5 \times 10^{-4}$
- **Augmentations:** Random resized crop to $224 \times 224$ with scale in $[0.9, 1.0]$ and aspect ratio in $[0.75, 1.333]$, using bicubic interpolation and antialiasing, followed by normalization with ImageNet mean and standard deviation.

This adaptation procedure was successfully applied to CLIP and MoCo v3 encoders. However, using the same LoRA configuration for DINOv2 led to unstable training and non-converging validation accuracy, even after extensive tuning of the learning rate, LoRA rank, and batch size. As such, we omit DINOv2+$L_i$ results.

**Classifier.** For all static pretraining experiments, we extract $\ell_2$-normalized image features using the frozen encoder—either with or without LoRA adaptation—and compute a nearest class mean (NCM) classifier for each train year. Prototypes (i.e., class means) are computed using the training set of that year, and evaluation is performed on each test year using all its labeled classes, regardless of whether those classes were present in the corresponding train year. This setup reflects the realistic challenge of generalizing to both previously seen and unseen classes across time. We use cosine distance to assign test samples to the class with the closest mean feature vector, a standard approach when working with normalized embeddings. Evaluation images are preprocessed with:

- **Validation transforms:** Resize to 224 pixels on the shorter side with bicubic interpolation, center crop to $224 \times 224$, then normalize using ImageNet mean and standard deviation.

## C.2 Additional static pretraining results

**DINOv2 train-test year performance.** Figure 5 presents the complete train-test accuracy matrices for DINOv2 ViT-S and ViT-B. We excluded these heatmaps from the main submission due to DINOv2's consistently low performance across all year splits, but include them here for completeness and transparency. DINOv2 exhibits substantially lower accuracy compared to all other pretrained models evaluated. To appropriately visualize this lower range of values, we employ a different color map and a separate color scale from that used in the main paper. This ensures that variation across train-test year pairs remains visible, despite the overall lower performance. The poor performance may also help explain the unstable training observed during LoRA adaptation. Unlike CLIP or MoCo v3, which produce embeddings that are more linearly separable for fine-grained car classes, DINOv2 appears less aligned with the class boundaries relevant to this task. As a result, lightweight adaptation methods such as LoRA may be insufficient to bridge this gap.

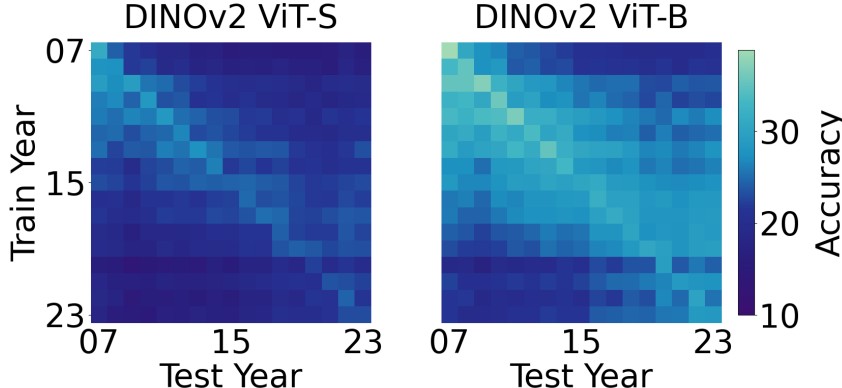

Figure 5: Full train-test accuracy matrix (%) for DINOv2 ViT-S and ViT-B. Performance is significantly below that of domain-specific or LoRA-adapted models.

**Statistical comparison of static pretraining variants.** To assess the robustness of the performance differences reported in Section 4.1, we conducted pairwise statistical comparisons across all models. For each model pair, we computed the difference between their 17×17 train-test accuracy matrices, flattened these into 289-element vectors, and applied a Wilcoxon signed-rank test. This test evaluates whether one model consistently outperforms another across the temporal grid. All pairwise differences were statistically significant ($p < 0.01$), except for the comparison between CLIP-B+LoRA and MoCo v3-B+LoRA, which yielded a p-value of 0.63. This suggests that the two models achieve relatively similar performance, with MoCo v3-B+LoRA holding a slight edge in average accuracy. Full test results are shown in Figure 6.

**Tradeoffs between CLIP adaptation and MoCo v3 pretraining.** Although the adapted MoCo v3 ViT-B slightly outperforms CLIP+$L_i$ in static accuracy, this gain comes with a significant training cost: MoCo v3 requires full in-domain pretraining on 1.9M car images, whereas CLIP achieves comparable results through lightweight LoRA adaptation using only the labeled 2007 subset. CLIP's

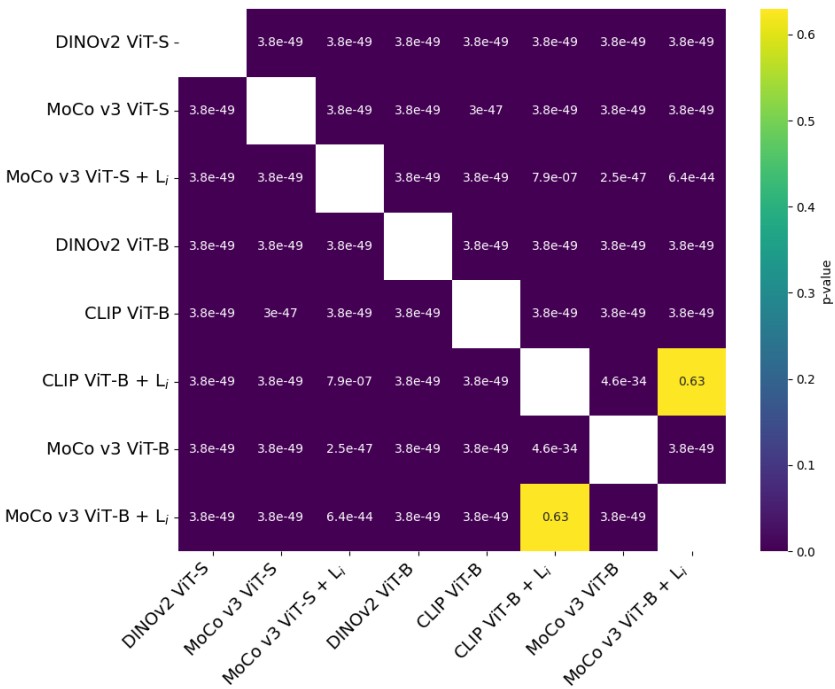

Figure 6: Wilcoxon signed-rank test results across all model pairs . All differences are statistically significant ($p < 0.01$) except CLIP-B+LoRA vs. MoCo v3-B+LoRA ($p = 0.63$)

advantage is partially attributable to its large-scale pretraining on LAION-2B, which includes image-text pairs collected up to 2021—introducing possible temporal overlap with CaMiT test years. Such overlap may inflate CLIP's performance on more recent data. Moreover, reproducing a CLIP-scale model using data strictly before 2007 is currently infeasible due to compute and data limitations. Despite these caveats, the strong performance of CLIP with minimal adaptation suggests that large-scale generic encoders can be competitive alternatives to domain-specific models in static scenarios. However, these conclusions are based on evaluations with a simple Nearest Class Mean (NCM) classifier, which may underutilize the full representational capacity of the encoders. As shown in our Time-Incremental Classifier Learning (TICL) experiments (Section 4.3), more expressive classifiers expose a growing performance gap: specialized models such as MoCo v3 with LoRA better consolidate temporal knowledge, while frozen encoders like CLIP plateau. This highlights the limits of general-purpose encoders under evolving data distributions and emphasizes the importance of considering both encoder quality and classifier capacity in continual learning.

## D  Time-incremental pretraining

### D.1  TIP Model Training Procedure

While Section 4.2 outlined our reservoir-based continual pretraining strategy, here we expand on the implementation details and justify our design choices in light of recent work on efficient continual learning, notably TIC-CLIP [6].

Our primary goal is to assess whether time-incremental pretraining (TIP) improves downstream performance compared to static pretraining, especially under temporal distribution shifts. To do so under a fixed compute and memory budget, we adopt a streamlined yet effective training protocol inspired by the cumulative-equal memory strategy introduced in TIC-CLIP. Instead of updating a single model incrementally, we train separate model instances for each year using temporally balanced datasets that integrate both current and historical data. This design preserves the core continual learning constraint—limited access to past data—while enabling clear, interpretable comparisons across time.

We begin with the 2005–2007 MoCo v3 ViT-S model used in the static pretraining (SPT) baseline. For each subsequent year $y \in Y = \{2008, \ldots, 2020\}$, we construct a 1.9 million image training set using a sequential memory accumulation scheme that mimics a time-aware reservoir buffer:

- We allocate 200,000 images from the current year $y$, using all available labeled and unlabeled data.
- We construct a replay memory of 1.7 million images from previous years ($< Y$), updated incrementally and sampled uniformly per year.

The 1.9M image budget per year is directly inherited from the static pretraining setup to ensure fair comparisons. In TIP, this budget is fulfilled via replay, preserving chronological diversity while maintaining parity with the initial static training cost.

The replay memory is assembled chronologically. For each prior year $y < Y$, we sample:

$$M = \min\left(200{,}000, \frac{1{,}700{,}000}{N}\right) \tag{6}$$

where $N$ is the number of previous years.

Equation 6 ensures temporal balance, prevents over-representation of any single year, and enforces a strict global budget. Importantly, the replay memory construction is consistent across years, making performance comparisons interpretable and fair.

This chronologically balanced scheme follows the Cumulative-Equal strategy in TIC-CLIP [6], which showed that simple temporal replay can match the performance of more complex memory strategies while requiring significantly less compute. In our context, it also facilitates direct evaluation of how TIP closes the generalization gap left by static pretraining.

A practical advantage of our approach is that, once the replay memory is constructed for each year, model training can proceed independently across time steps. This enables all yearly TIP models to be trained in parallel, significantly reducing wall-clock time and making large-scale temporal evaluation feasible under fixed resource constraints. Such parallelism is especially valuable in long-horizon settings like ours, where evaluating across many years is critical for understanding the effects of temporal shifts.

**Variants of Time-Incremental Pretraining.** We evaluate three TIP variants, all designed to improve over static pretraining by leveraging temporal supervision. These variants differ in how they incorporate past data—via full retraining or adapter tuning—but share the same underlying goal: increasing robustness to time-evolving data.

- **Reservoir-Based TIP (Full Model Retraining, R):** For each year $Y$, we train a new model from scratch using 1.9M images composed of 200K current-year samples and 1.7M temporally balanced samples from earlier years, as described above. This serves as our default approach and provides a strong baseline for evaluating the benefits of replay in a controlled, time-aware setting.

- **Sequential LoRA Adaptation ($L_c$):** Starting from the static 2005–2007 model, we fine-tune LoRA adapters year by year using only the *labeled data from the current year*. LoRA weights are updated incrementally, allowing the model to adapt with minimal additional compute. This variant reflects a practical, low-cost way to continually adapt a model over time, and corresponds to '$L_c$' in Table 2 of the main submission.

- **Reservoir-Based LoRA Adaptation ($L_a$):** In this hybrid approach, we apply LoRA fine-tuning on top of the frozen backbone trained using reservoir-based TIP. For each year $Y$, LoRA adapters are updated using all *labeled data from years $\leq Y$*. Unlike the sequential version, this avoids weight accumulation and allows us to evaluate LoRA under the same memory constraints as full retraining. This variant corresponds to '$L_a$' in Table 2.

While these strategies differ in computational and memory tradeoffs, they all significantly outperform static pretraining (Table 2 of the main text), demonstrating the value of TIP in mitigating temporal degradation. Sequential LoRA offers an efficient, real-world adaptation pathway, while reservoir-based variants serve as a controlled baseline to study the long-term effects of data replay.

**Training Details.** For reservoir-based TIP models, we initialize from the static MoCo v3 ViT-S model (trained on 2005–2007) and train each yearly instance from scratch using the replay buffer. We follow the same optimization settings as in static pretraining, but reduce the base learning rate to $1 \times 10^{-4}$ and train for 60 epochs to account for the model's prior pretraining and the moderate adaptation required each year.

For LoRA-based variants, we fine-tune only the adapter weights using labeled data. Each yearly update is performed for 20 epochs with a batch size of 48, using SGD with momentum 0.9, weight decay 0.0005, and a cosine annealing learning rate scheduler with a base learning rate of 0.01. These settings reflect the lightweight nature of LoRA adaptation and are chosen to ensure stable incremental updates with limited compute.

## D.2 Additional TIP Results

Due to time and compute constraints, we were unable to complete full model retraining or reservoir-based LoRA adaptation for CLIP and MoCo v3 ViT-B. However, we provide results for sequential LoRA adaptation on both models. These results show that even lightweight adapter tuning offers clear performance gains over static pretraining in the presence of temporal distribution shifts for larger models like ViT-B (see Figure 7).

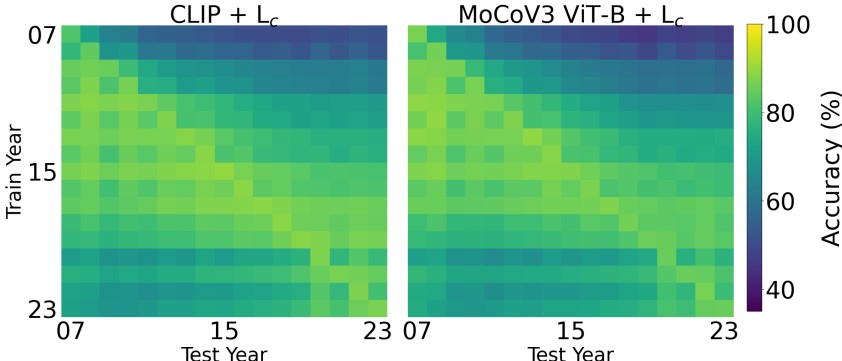

Figure 7: Train-test accuracy matrix (%) for CLIP ViT-B and MoCo v3 ViT-B across years. TIP via sequential LoRA adaptation leads to consistent gains over static pretraining, improving robustness to forward and backward temporal shifts

To quantify the performance gains of TIP over static pretraining, we performed pairwise statistical comparisons between each TIP variant and its corresponding static baseline. For each model pair, we computed the difference between their 17×17 train-test accuracy matrices, flattened these into 289-element vectors, and applied a Wilcoxon signed-rank test. All comparisons yielded statistically significant improvements in favor of TIP ($p < 0.01$), indicating that TIP models consistently

outperform static models across temporal evaluation points. These results highlight the robustness and generalizability of TIP's temporal adaptation. Full statistical test results are shown in Figure 8.

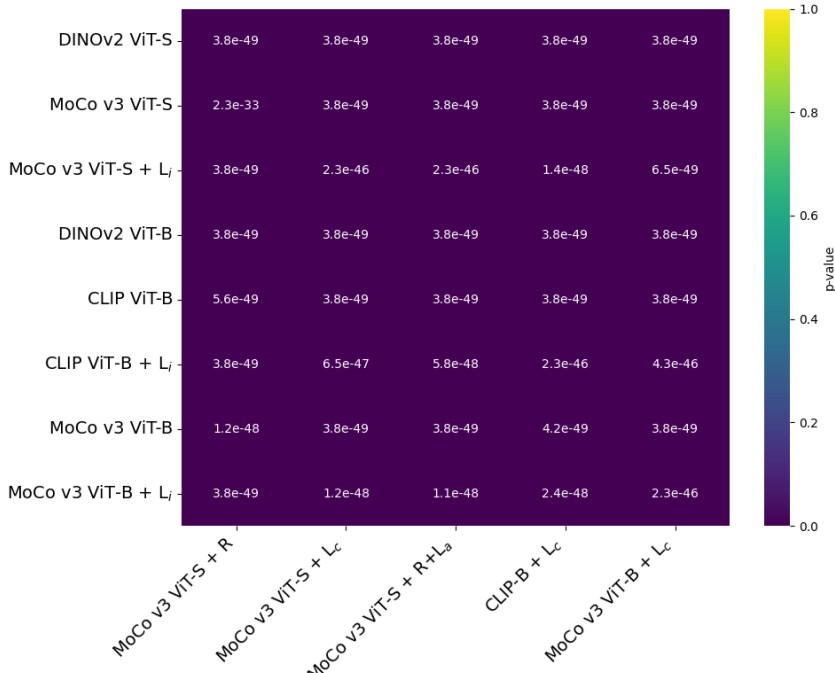

Figure 8: Statistical test results comparing TIP and static pretraining variants. Each cell shows the $p$-value from a Wilcoxon signed-rank test comparing one TIP model (columns) to a static model (rows) over the full 17×17 temporal accuracy grid. All comparisons are significant at $p < 0.01$.

**TIP variant comparison.** To support the findings in Section 4.2 of the main paper, we report statistical comparisons between the TIP variants—reservoir-only (R), sequential LoRA (Lc), and combined LoRA with reservoir (R+La)—across ViT-S and ViT-B backbones. As shown in Figure 9, nearly all pairwise differences are statistically significant ($p < 0.01$), with the exception of MoCo v3 ViT-S + Lc versus CLIP ViT-B + Lc ($p = 0.67$). Interestingly, MoCo v3 ViT-S + Lc slightly outperforms MoCo v3 ViT-B + Lc across all TIP metrics, suggesting that higher model capacity does not necessarily yield better temporal generalization under lightweight adaptation. These results complement our static pretraining analysis (Appendix C), where CLIP performed well with limited adaptation. In the TIP setting, its strong performance persists under sequential LoRA. However, the use of simple Nearest Class Mean classifiers may underutilize the full representational capacity of different encoders, compressing performance differences. As we explore in Appendix E, these differences become more pronounced when more expressive classifiers are introduced in the TICL setting—highlighting the limitations of frozen, generic models under long-term temporal shift.

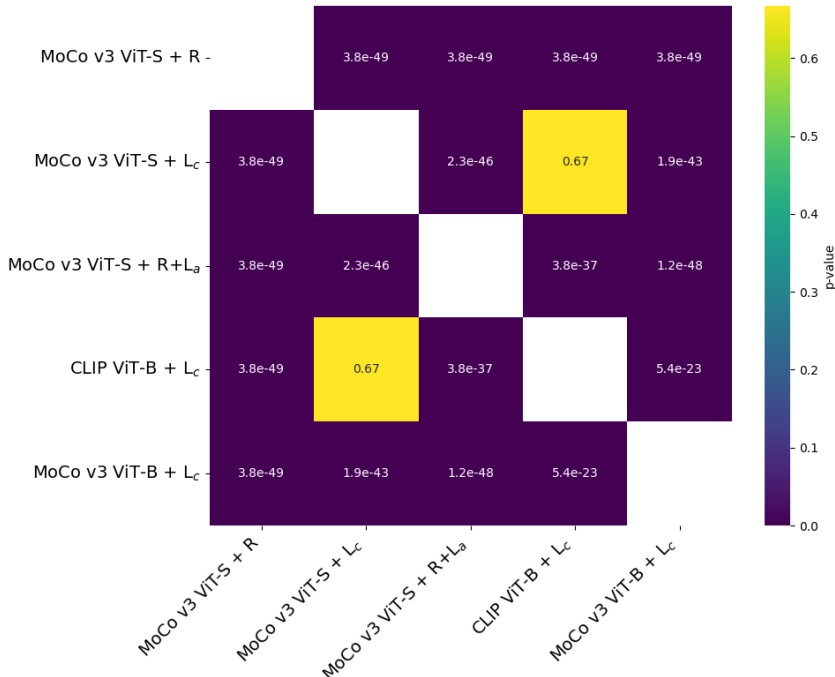

Figure 9: Statistical test results comparing TIP variants. Each cell shows the $p$-value from a Wilcoxon signed-rank test comparing two TIP models (R: reservoir-only, Lc: LoRA-only, R+La: combined) over the flattened $17 \times 17$ accuracy matrix. All differences are significant at $p < 0.01$.

# E   Time-incremental classifier learning

## E.1   TICL Methods: Training Procedure

In Section 4.3, we experimented with Time-Incremental Classifier Learning (TICL) as a lightweight and effective strategy to adapt classifiers over time without retraining the backbone. Unlike Time-Incremental Pretraining (TIP), TICL freezes the feature extractor and updates only the classifier incrementally as new data becomes available each year. This modular approach reduces computational cost and memory usage by freezing the encoder, while enabling incremental classifier adaptation to evolving data distributions. Below, we detail the implementation of the four TICL methods evaluated in our experiments: NCM-TI, FeCAM, RanDumb, and RanPAC.

**NCM-TI** This method extends the standard Nearest Class Mean (NCM) classifier to handle incremental data arrival. Each class $c$ is represented by a prototype $\mu_c \in \mathbb{R}^d$ computed as the mean of its feature embeddings. In the time-incremental setting, these prototypes are updated proportionally with each year's new samples. Let $f(x)$ denote the frozen encoder output, $n_{\text{old}}$ the number of accumulated samples up to year $y-1$, and $n_{\text{new}}$ the number of new samples in year $y$. Then:

$$\mu_c^{(y)} = \frac{n_{\text{old}} \cdot \mu_c^{(y-1)} + \sum_{i=1}^{n_{\text{new}}} f(x_i)}{n_{\text{old}} + n_{\text{new}}} \tag{7}$$

Equation 7 defines the incremental update rule for NCM-TI. Inference is performed using the cosine similarity between a test feature $f(x_{\text{test}})$ and each class prototype. This method requires no storage of past data or per-class covariance, ensuring low overhead. As in the original NCM, no second-order statistics are maintained. For $C = 190$ and $d = 768$, this method adds 146K paramters, with one 768-dimensional prototype per class.

**FeCAM** FeCAM enhances NCM by modeling class-wise uncertainty via covariance matrices. For each class $c$, it maintains both a prototype $\mu_c \in \mathbb{R}^d$ and a second-order moment matrix $S_c \in \mathbb{R}^{d \times d}$. They are updated incrementally using linear weighting based on sample counts. Let $\mu_c^{(y-1)}$ and $S_c^{(y-1)}$ be the previous estimates:

$$\mu_c^{(y)} = \frac{n_{\text{old}} \cdot \mu_c^{(y-1)} + \sum_{i=1}^{n_{\text{new}}} f(x_i)}{n_{\text{old}} + n_{\text{new}}} \tag{8}$$

$$S_c^{(y)} = \frac{n_{\text{old}} \cdot S_c^{(y-1)} + \sum_{i=1}^{n_{\text{new}}} f(x_i)f(x_i)^\top}{n_{\text{old}} + n_{\text{new}}} \tag{9}$$

The empirical class covariance is then computed as:

$$\Sigma_c^{(y)} = S_c^{(y)} - \mu_c^{(y)}\mu_c^{(y)\top} \tag{10}$$

Equations 8, 9, and 10 jointly define FeCAM's update rule. Inference is performed via Mahalanobis distance using $\mu_c^{(y)}$ and $\Sigma_c^{(y)}$. This approach better captures class distributions at the cost of storing a covariance matrix per class. For $C = 190$ and $d = 768$, FeCAM adds 112.6M parameters (146K for class means and 112.5M for class covariances).

**RanDumb** RanDumb projects features into a high-dimensional space using a fixed nonlinear mapping via a random RBF kernel. Here, $\phi_{\text{RBF}}(\cdot)$ denotes a fixed random feature approximation of an RBF kernel. Let $f(x) \in \mathbb{R}^d$ be the frozen backbone output and $\phi_{\text{RBF}}(f(x)) \in \mathbb{R}^r$ the projected feature, where $r = 10{,}000$. Each class maintains a prototype $\mu_c \in \mathbb{R}^r$, updated incrementally over time:

$$\mu_c^{(y)} = \frac{n_{\text{old}} \cdot \mu_c^{(y-1)} + \sum_{i=1}^{n_{\text{new}}, y_i = c} \phi_{\text{RBF}}(f(x_i))}{n_{\text{old}} + n_{\text{new}}} \tag{11}$$

To capture feature variability, a shared second-order moment matrix $S \in \mathbb{R}^{r \times r}$ is maintained across all classes:

$$S^{(y)} = \frac{n_{\text{old}} \cdot S^{(y-1)} + \sum_{i=1}^{n_{\text{new}}} \phi_{\text{RBF}}(f(x_i))\phi_{\text{RBF}}(f(x_i))^\top}{n_{\text{old}} + n_{\text{new}}} \tag{12}$$

The shared covariance matrix $\Sigma$ is then computed as:

$$\Sigma^{(y)} = S^{(y)} - \mu_{\text{all}}^{(y)}\mu_{\text{all}}^{(y)\top} \tag{13}$$

where $\mu_{\text{all}}^{(y)}$ is the global mean of all projected training samples. Equation 11 defines the class prototype update, Equation 12 defines the shared second-order statistics, and Equation 13 computes the shared covariance matrix. The empirical covariance is regularized using Oracle Approximating Shrinkage (OAS) for stability. Inference is performed with Mahalanobis distance using $\Sigma^{(y)}$. For $C = 190$ and $r = 10,000$, this method adds 101.9M parameters (1.9M for class prototypes and 100M for the shared covariance).

**RanPAC** RanPAC performs classification via regression in a high-dimensional random feature space. Let $\phi(f(x)) = Wf(x) \in \mathbb{R}^r$ denote a linear projection with $W \in \mathbb{R}^{r \times d}$ randomly initialized and fixed, and $r = 10,000$. For each sample $x_{y,n}$ in year $y$ with one-hot label $y_{y,n}$, let $h_{y,n} = \phi(f(x_{y,n}))$. The algorithm accumulates the matrices:

$$G = \sum_{y=1}^{Y}\sum_{n=1}^{N_y} h_{y,n}h_{y,n}^\top \tag{14}$$

$$C = \sum_{y=1}^{Y}\sum_{n=1}^{N_y} h_{y,n}y_{y,n}^\top \tag{15}$$

At inference, given a test instance $x_{\text{test}}$, the class score vector is computed as:

$$s = \phi(f(x_{\text{test}}))^\top (G + \lambda I)^{-1}C \tag{16}$$

where $\lambda$ is a regularization parameter selected via cross-validation. Equations 14, 15, and 16 fully define RanPAC's training and inference logic. For $r = 10,000$ and $C = 190$, RanPAC adds 101.9M parameters (100M for matrix $G$ and 1.9M for matrix $C$).

## E.2 Additional TICL Results

We present the full time-incremental accuracy matrices for all five TICL methods across the eight evaluated backbones. Each heatmap is a $17 \times 17$ matrix showing classifier accuracy when training on one year and testing on another. Rows represent training years and columns represent test years. These visualizations provide insights into both forward and backward transfer over time.

Figures 10 and 11 show the results for all eight backbones. Each figure contains four rows, where each row corresponds to a different backbone. Within each row, the five methods NCM, NCM-TI, FeCAM, RanDumb, and RanPAC are shown as adjacent heatmaps. This layout highlights how different backbones affect temporal generalization behavior.

**Statistical comparisons.** To support the observations made in Section 4.3, we conducted statistical significance tests for both TIP and TICL configurations under time-incremental evaluation. Figure 12 compares the performance of all TIP variants (R, Lc, R+La), trained from the static MoCo v3 ViT-S backbone, against TICL methods (NCM, NCM-TI, FeCAM, RanDumb, RanPAC), evaluated using the same backbone but with an initial LoRA adaptation trained on 2007 data. All comparisons are statistically significant ($p < 0.01$), indicating that the TICL methods consistently outperform TIP variants in this setting. This suggests that advanced classifier updates can more effectively exploit representational capacity than lightweight encoder updates alone.

Separately, Figure 13 reports pairwise statistical tests across TICL methods using the same MoCo v3 encoder. All differences are statistically significant at $p < 0.05$, with most well below $p < 0.01$.

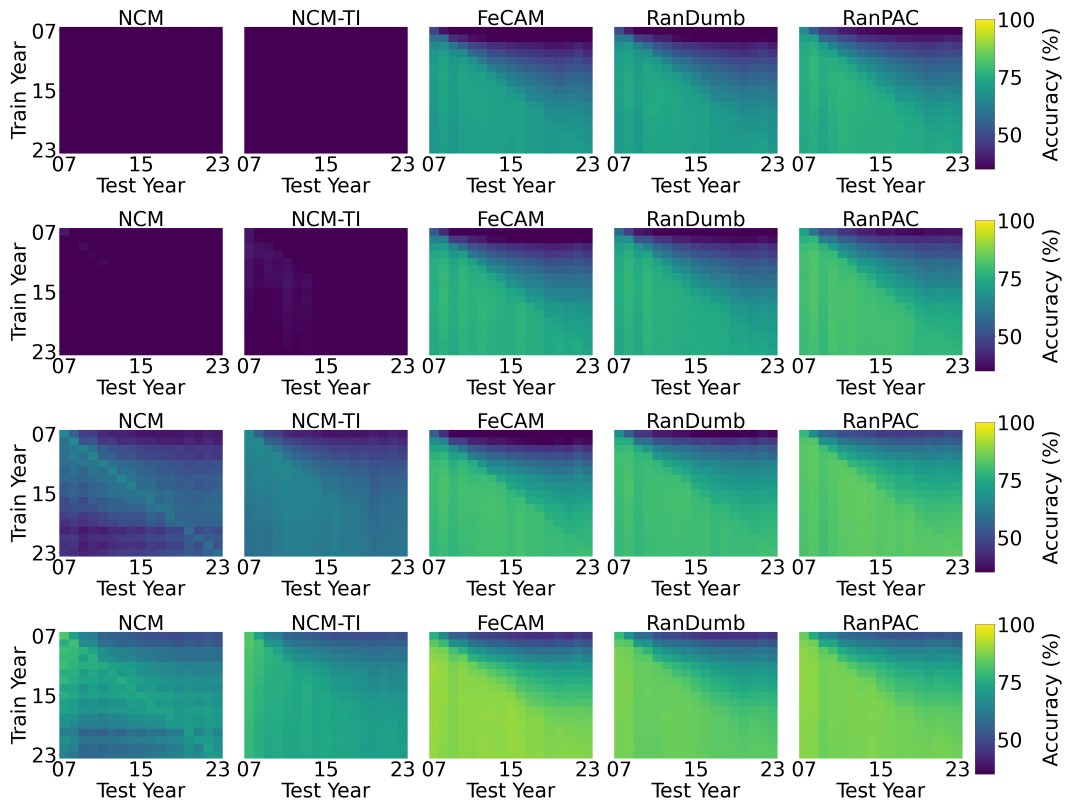

Figure 10: TICL accuracy matrices for (a) DINOv2 ViT-S, (b) DINOv2 ViT-B, (c) CLIP ViT-B, (d) CLIP ViT-B with LoRA.

The closest result occurs between FeCAM and RanPAC ($p = 0.019$), indicating that while RanPAC typically leads in accuracy, its advantage over FeCAM is modest. Overall, these results confirm that more expressive classifiers—particularly RanPAC, FeCAM, and RanDumb—consistently outperform simpler baselines such as NCM and NCM-TI under long-term incremental evaluation.

To further analyze generalization under time-incremental settings, we compare **CLIP ViT-B +** $L_i$ and **MoCo v3 ViT-B +** $L_i$ using various TICL methods. As shown in Figure 14, the difference between the two models is not statistically significant under the NCM classifier ($p = 0.63$)—as previously observed in the static pretraining analysis (Appendix C). However, when paired with more expressive TICL methods such as FeCAM, RanDumb, and RanPAC, MoCo v3 + $L_i$ consistently outperforms CLIP + $L_i$, with statistically significant differences ($p < 0.01$). These methods more effectively exploit the structure of learned features, revealing the added benefit of domain-specific pretraining. While CLIP continues to perform robustly, these results demonstrate that specialized encoders—when combined with lightweight tuning—offer superior long-term generalization when classification complexity increases. This highlights the importance of jointly considering encoder specialization and classifier expressiveness for sustained performance under temporal shift.

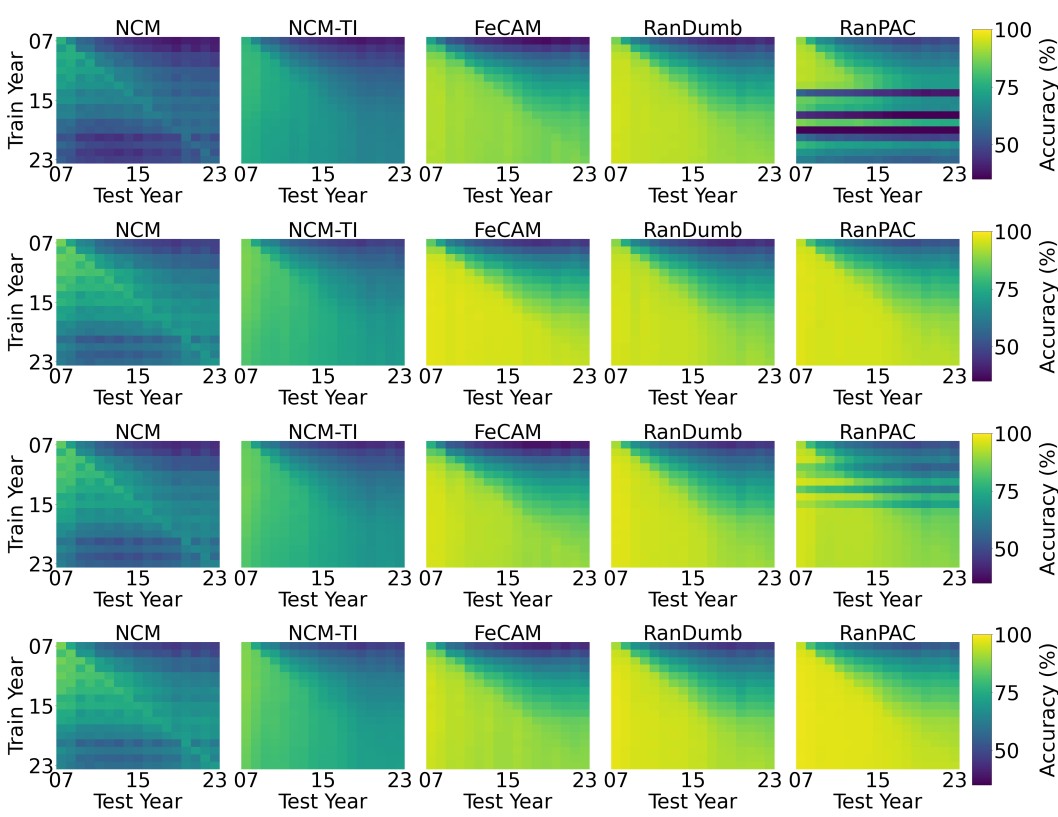

Figure 11: TICL accuracy matrices for (e) MoCo v3 ViT-S, (f) MoCo v3 ViT-S with LoRA, (g) MoCo v3 ViT-B, (h) MoCo v3 ViT-B with LoRA.

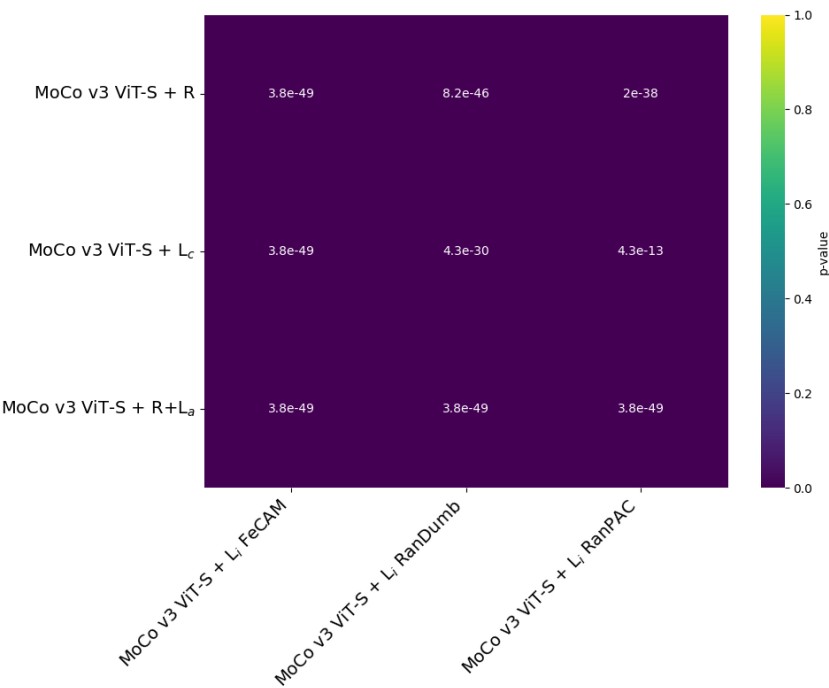

Figure 12: Statistical test results comparing TIP variants (R, Lc, R+La) against TICL methods (NCM, NCM-TI, FeCAM, RanDumb, RanPAC) using MoCo v3 ViT-S with initial LoRA adaptation. Each cell shows the $p$-value from a Wilcoxon signed-rank test based on the $17{\times}17$ accuracy matrix. All comparisons are significant at $p < 0.01$.

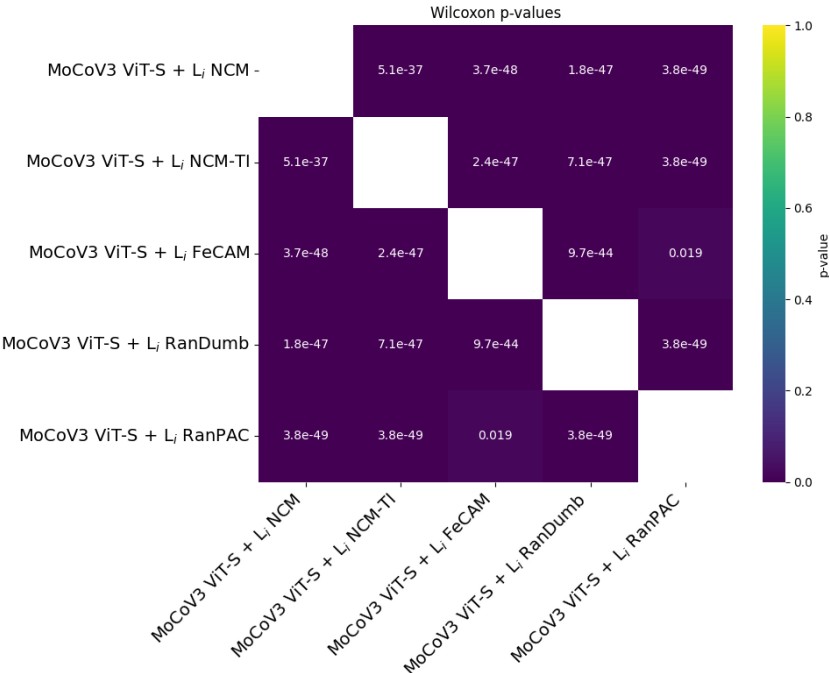

Figure 13: Statistical test results comparing TICL methods (NCM, NCM-TI, FeCAM, RanDumb, RanPAC) using MoCo v3 ViT-S with initial LoRA adaptation. Each cell shows the $p$-value from a Wilcoxon signed-rank test based on the $17{\times}17$ accuracy matrix. All comparisons are significant at $p < 0.01$.

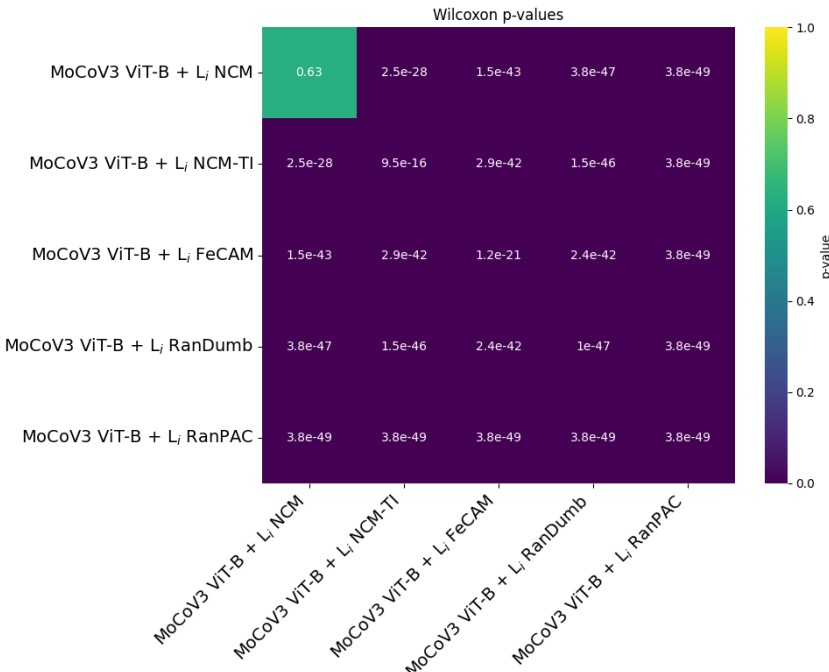

Figure 14: Statistical test results comparing CLIP ViT-B and MoCo v3 ViT-B with LoRA adaptation across different TICL methods. Each cell shows the $p$-value from a Wilcoxon signed-rank test based on the $17{\times}17$ accuracy matrix. While the difference is not significant under NCM ($p = 0.63$), it becomes highly significant ($p < 0.01$) with more expressive classifiers such as FeCAM, RanDumb, and RanPAC.

## F Time-aware image generation

### F.1 Generative model fine-tuning procedure

We implemented a custom dataset that loads 655,681 image-caption pairs. For the $\text{SD1.5}_{\text{T AIG}}$ model, each caption is formatted as "A photo of a {car_model} in {year}", with images organized into corresponding car class and year folders. For the $\text{SD1.5}_{FT}$ model, captions follow the format "A photo of a {car_model}", with images grouped by car class folders. Both models were trained for 30 epochs with a batch size of 64, totaling 307,380 optimization steps. So $\text{SD1.5}_{\text{T AIG}}$ learn temporal variations for year-specific generation, while $\text{SD1.5}_{FT}$ focuses on general car model representation.

Our models use pretrained Stable Diffusion 1.5 [23] components: a frozen VAE, UNet, and CLIP text encoder. Only LoRA adapters are trained, which are inserted into the UNet's attention layers, targeting the key, query, value, and output projection modules. This adds approximately 12.75 million trainable LoRA parameters, roughly 1.4% of the full Stable Diffusion 1.5 backbone.

Our training follows the diffusion framework: input images are first encoded into a latent representation by the frozen VAE. Gaussian noise is then added to these latents according to randomly sampled timesteps, simulating the forward diffusion process. The UNet takes the noisy latents along with corresponding text embeddings from the frozen CLIP text encoder and predicts the noise component at each timestep. The model is trained to minimize the mean squared error (MSE) between its predicted noise and the true noise added, which can be optionally reweighted by a signal-to-noise ratio (SNR) factor to emphasize learning on certain noise levels.

Optimization uses AdamW with a constant learning rate of $1 \times 10^{-4}$, batch size 64 and mixed precision = `bf16`. Training is managed via the Hugging Face Accelerate library [9], with gradient accumulation and clipping settings to ensure stable distributed training.

### F.2 Visualizing Temporal Dynamics in Generated Samples

We evaluate whether the generated images reflect temporal variation present in the corresponding real image distributions. We focused and generated models with the highest class dynamics ranked using real image distributions and checked whether their generative counterparts show similar temporal shifts. In comparison, the zero-shot generations lacked temporal differentiation, often repeating similar visual features regardless of the input year. Below, we show results for two high-dynamics classes: Ford Taurus (Figure 15) and Dodge Durango (Figure 16)

**Ford Taurus**

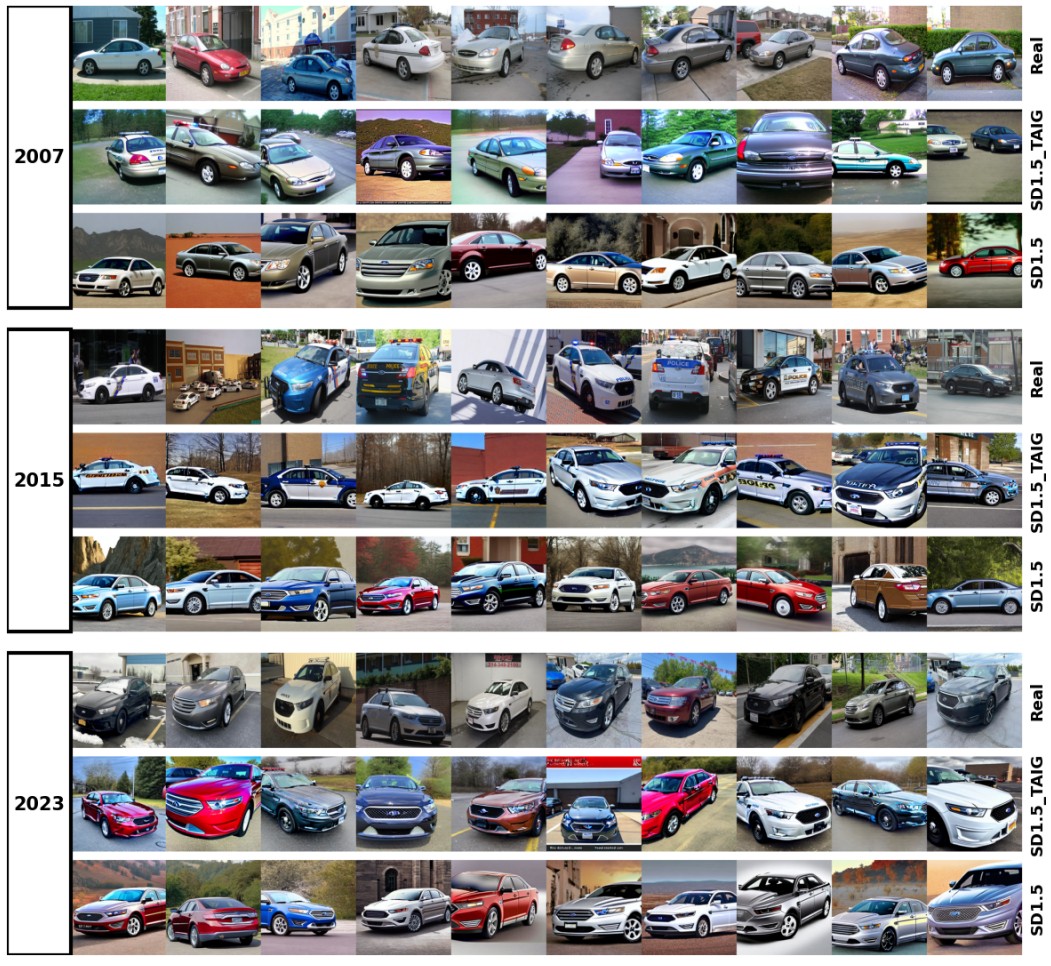

Figure 15: Temporal comparison of visual changes for the **Ford Taurus** class, arranged in a 9-row by 10-column grid. Rows are grouped by year (2007, 2015, 2023) and data source: real images (rows 1, 4, 7), fine-tuned SD1.5$_{\text{T AIG}}$ generations (rows 2, 5, 8), and zero-shot generations (rows 3, 6, 9). Each cell contains one of 10 image samples per row. Real images show distinct changes over time. Finetuned generations from SD1.5$_{\text{T AIG}}$ also capture these temporal trends. In contrast, zero-shot generations appear largely invariant to year conditioning, often repeating similar images across time.

**Dodge Durango**

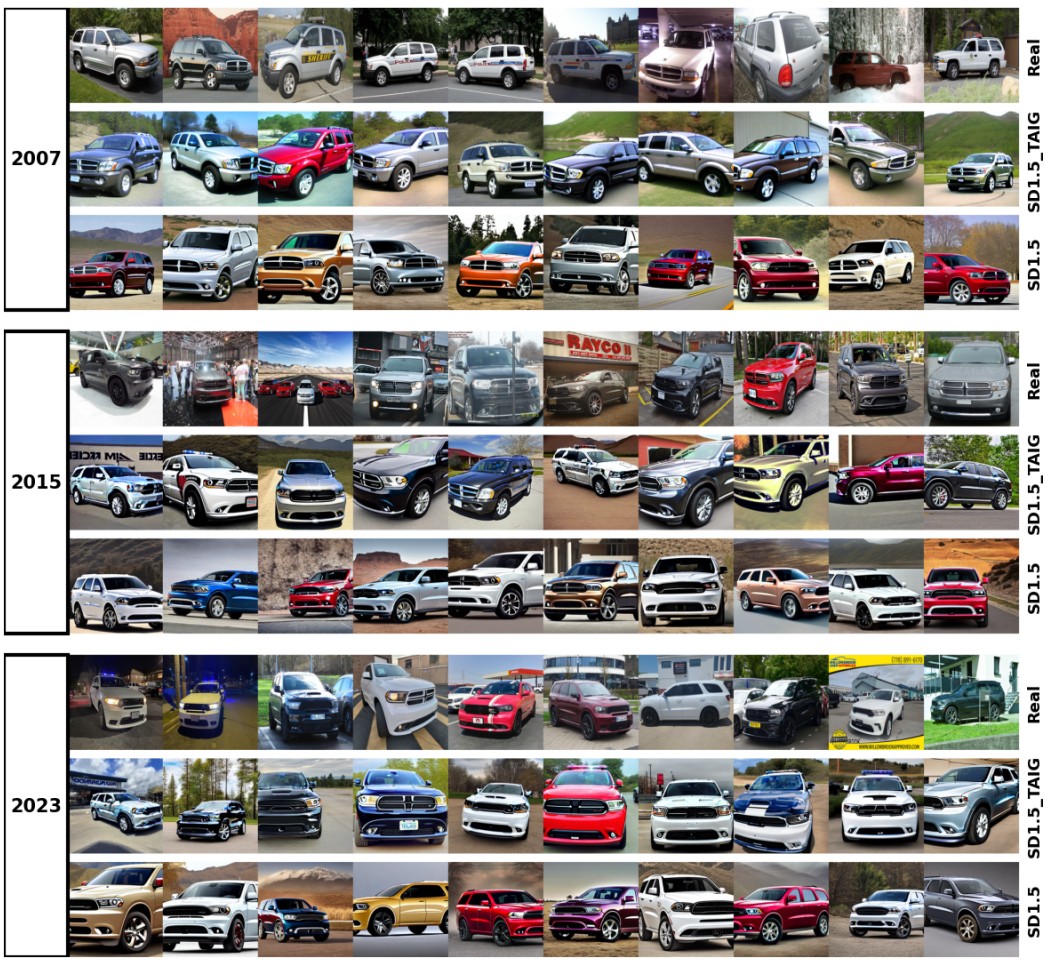

Figure 16: Temporal comparison of visual changes for the **Dodge Durango** class, arranged in a 9-row by 10-column grid. Rows are grouped by year (2007, 2015, 2023) and data source: real images (rows 1, 4, 7), fine-tuned $SD1.5_{T\ AIG}$ generations (rows 2, 5, 8), and zero-shot generations (rows 3, 6, 9). While real images show clear shifts over time, only the $SD1.5_{T\ AIG}$ generations reflect these changes. Zero-shot outputs again fail to capture temporal variation, often producing similar outputs across years.

## G  Experiment Costs

### G.1  GPU Hours

In total, we estimate that approximately **1300 A100-hours** were used to run the experiments presented in the main paper. This includes model pretraining, time-incremental learning, classifier evaluations, and image generation. Most experiments were conducted on local infrastructure, sometimes using multi-GPU setups.

In addition to the GPU-hours estimation reported above, we spent approximately **4100 A100-hours** for dataset creation. The bulk of this compute was consumed by Qwen-based vision-language inference used for large-scale filtering and annotation. We cannot reliably estimate the GPU-hours required for ChatGPT-based annotation, but we can safely assume they are similar or higher than those of Qwen. With this assumption, the total number of GPU-hours needed to construct the dataset amounts to approximately **9300 A100-hours or more**.

### G.2  Monetary Cost

The estimated monetary cost for GPT-based annotation and filtering is approximately **$638.83**, based on GPT API usage. All GPT queries were submitted via the batch API, which reduces costs by half compared to unbatched inference.

Additionally, manual annotation was conducted over **56.1 hours** and compensated at an hourly rate of **$12**, resulting in a total labor cost of **$673.20**.