# OpenReview forum: "CaMiT: A Time-Aware Car Model Dataset for Classification and Generation"
_NeurIPS.cc/2025/Datasets_and_Benchmarks_Track — NeurIPS 2025 Datasets and Benchmarks Track poster_

### Official Review · Reviewer_kcf5 · 2025-06-30

**Rating:** 4
**Confidence:** 3

**Summary:**

This paper creates a new benchmark for evaluting the performance of models in classification and generation tasks. Specifically, thi paper introduces Car Models in Time (CaMiT). This fine-grained dataset is used to capture the temporal evolution of this representative subset of technological artifacts. CaMiT includes 787K labeled samples of 190 car models (2007–2023) and 5.1M unlabeled samples (2005–2023), supporting supervised and self-supervised learning. This paper also investigates two mitigation strategies: time-incremental pretraining and time-incremental classifier learning and provide good expeirment results.

**Dataset Code Accessibility:**

Yes

**Ethical Considerations:**

No, there are no or only very minor ethics concerns

**Final Justification:**

We have read the author's responses. My primary concenrs, including the comparison between the proposed dataset and other datasets, the experiments and the time-incremental classification. Based on the author's feedbacks and comments from other reviewers, I recommend borderline accept.

**Limitations Weaknesses:**

One of the primary weakness of this paper is that there are many datasets such as Air Quality and Human Activity Recognition Using Smartphones [1] for the time-series applications and the discussion between this paper and other related works should be enhanced.

[1] Anguita, Davide, et al. "A public domain dataset for human activity recognition using smartphones." Esann. Vol. 3. No. 1. 2013.

The other weakness of this paper is that the expeirments are mainly used for the time-series classificaiton. Can the proposed dataset be applied to other classification tasks?

In addition, this paper focuses on the time-incremental classification. The connection between the the time-incremental classification and the real-world applications should be further explained.

Furthermore, the experiments are performed when using the Vision Transformer (ViT) as the backbone. We would like to see the performance of other backbones such as the ResnetNet. We also encourage authors to compare more continual learning baselines that were published in CVPR2025.

**Strengths Contributions:**

1. The dataset intrdouced in this paper has values such as the the sample number of the dataset for the computer science field. In addition, this dataset also supports different model training configurations such as the unsupervised and semi-supervised learning.

2. The writting of this paper is good and the motivation is clearly described.

3. This paper constructs a series of experiments to demonstrate the effectiness of the dataset.

---

> ### Author Rebuttal · Authors · 2025-07-29
>
> Thank you for your careful review and for emphasizing CaMit’s large scale support for both supervised and self-supervised learning, clear motivation and comprehensive experiments.
>
>  Below, we address each of your suggestions :
>
> 1. Related-work framing: Time handling in AI models and systems is a multifaceted problem, driven, among others, by the definition of temporal granularity, data modalities, evaluated tasks, and data size. The related datasets in Section 2 of the submission focus on particular aspects of time modeling as does the suggested reference [1]. This dataset is indeed relevant prior work, and we will add it in the discussion of prior work.
>
>
> 2. Applicability beyond classification : Although our experiments center on classification and generation, CaMiT’s timestamped images naturally enable other tasks such as few-shot cross-year retrieval (identifying a model given only an older photo) and domain adaptation across time gaps. We also provide the bounding boxes of the car instances to enable temporal object detection. Similarly, time-aware instance segmentation is doable by applying SAM to the cropped bounding boxes.
>
> 3. Real-world relevance of time-incremental classification :
> Time-incremental classification instantiates the following question "How should visual domains whose appearance evolves over time be handled by AI models ? ", itself included in the more general question about novelty handling in AI models that motivates continual learning as a whole. Such questions are important in practice because AI models should be able to ingest new data while preserving past knowledge, a challenging trade-off. The TIP and TICL scenarios tested in experiments provide complementary ways to address this trade-off. In practice, a production-level model should be able to integrate very recent content and use it for making predictions.
>
> 4. Backbone diversity :
> We focus on ViT for experiments because this type of backbone is majoritarily used in recent continual learning papers for their effectiveness. We agree that testing with other backbones such as ResNet, contributes to the comprehensiveness of the conclusions and provide supplementary SPT and TICL experiments with this backbone. More specifically, we ran a MoCo v3 – ResNet 50 pretraining for 200 epochs and tested the TICL methods with this backbone and we observe the same temporal effect as with ViTs. We also observe that the performance of the ResNet backbone is lower than ViTs as we can expect.
> We report the following average accuracies for the TICL experiments with MoCo v3 – ResNet 50 :
>
>         –  NCM: 46.0%
>
>         –  NCM-TI: 49.4%
>
>         –  FeCAM: 57.1%
>
>         –  RanDumb:  53.2%
>
>         –  RanPAC: 59.3%
>
> 5. CVPR baselines: We surveyed the latest CVPR2025 literature, identifying adapter-based approaches like [2] and [3] that could fit into the TIP section of our paper. We also identified [4] that could fit into the TICL section of our paper, but these papers appeared after our submission deadline. They could not be fully implemented within the one-week rebuttal window.
>
> [1] Anguita, Davide, et al. "A public domain dataset for human activity recognition using smartphones." Esann. Vol. 3. No. 1. 2013.
>
> [2] He Jiangpeng, Duan Zhihao and Zhu Fengqing. "CL-LoRA: Continual Low-Rank Adaptation for Rehearsal-Free Class-Incremental Learning". CVPR, 2025.
>
> [3] Takuma Fukuda, Hiroshi Kera and Kazuhiko Kawamoto. “Adapter Merging with Centroid Prototype Mapping for Scalable Class-Incremental Learning”. CVPR, 2025
>
> [4] Yusong Hu, Zichen Liang, Fei Yang, Qibin Hou, Xialei Liu and Ming-ming Cheng. “KAC: Kolmogorov-Arnold Classifier for Continual Learning”. CVPR, 2025

---

> > ### Comment · Reviewer_kcf5 · 2025-08-06
> > **Official comments by reviewers**
> >
> > We thanks for the author's feedbacks, which address most of my main concerns. Therefore, I will maintain my original score (4).

---

> > > ### Author Response · Authors · 2025-08-06
> > >
> > > Thank you for this positive comment and acknowledging that most of the main concerns were addressed in the rebuttal!

---

### Official Review · Reviewer_eyUP · 2025-06-30

**Ethics Flags:** Data privacy, copyright, and consent
**Rating:** 5
**Confidence:** 4

**Summary:**

This work proposes a temporal car dataset CaMiT for image classification and generation tasks. The billion-scale dataset is collected, filtered, and annotated in the proposed fully/semi-automatic way. For the image classification task, the submission introduces three model training strategies: static pre-training (SPT), time-incremental pre-training (TIP), and time-incremental classifier learning (TICL), and provides a detailed experimental analysis. For the time-aware image generation task, the experimental results on time-aware fine-tuning demonstrated the effectiveness of this dataset in improving generation performance.

**Dataset Code Accessibility:**

Yes

**Dataset Code Comments:**

The code and reproducing instructions are released in a public GitHub repository. The dataset is released in a public HuggingFace repository.

**Ethical Comments:**

The submission complies with copyright regulations by distributing image links, embeddings, and metadata, but not the images themselves.

**Ethical Considerations:**

No, there are no or only very minor ethics concerns

**Final Justification:**

In the rebuttal, authors mention some methods to address the data imbalance issue and plan to annotate the dataset with more detailed labels, including appearance change and design modifications. My concerns are addressed, and I maintain my current score to accept this submission.

**Limitations Weaknesses:**

- I would like to raise a related question that is not discussed in the submission. When people talk about time-awareness, it more likely refers to an identical item at different timestamp. For example, if I purchased a C3 I in 2002, what would it look like in 2010? In Section 4.4 Time-aware image generation, given two prompts with the same car model but different timestamps (Prompt 1: “A photo of a C3 I in 2002” and Prompt 2: “A photo of a C3 I in 2010”), what is the difference between the two outputs of the time-aware diffusion model? I think an ideal expectation is that the former output should be a brand-new C3 I, while the latter should be a C3 I with a worn appearance. I am wondering whether CaMiT and the proposed TAIG diffusion model can handle this task.
- As the submission mentioned in Section 5 Limitations, there exists the extremely heavy data imbalance problem. I am wondering whether the authors consider potential strategies to alleviate the data imbalance problem.

**Strengths Contributions:**

- The submission proposes fully/semi-automatic methods to process the billion-scale dataset by leveraging the Flickr API to collect the data, YOLO, Qwen, and SAM models to filter it, and  Qwen, ChatGPT, MoCo models to annotate it.
- The depiction of the dataset is clear, with statistical tables and demonstration figures.
- Comprehensive experiments and analyses are conducted on the dataset via SPT, TIP, and TICL for time-aware image classification, and FT and TAIG for time-aware image generation.

---

> ### Author Rebuttal · Authors · 2025-07-29
>
> Thank you for your insightful review and for highlighting the clarity of our dataset depiction and the comprehensiveness of our experiments.
>
> Below, we address your comments:
>
> 1.Time-aware generation : CaMiT uses each image’s Flickr publish year, which captures both genuine design-variant introduction and a nostalgia effect (users sometimes upload older model photos well after their release). As a result, our time aware generation outputs mix these two phenomena : you’ll see facelifted model designs when a new variant appears and occasional "aged" uploads from nostalgic posts. To disentangle physical aging from design evolution, future works could align timestamps with official release dates and metadata (first registration of a model) enabling separate modeling of car usage and variant style effects. To integrate different effects, the task could be handled by envisioning it as a set generation rather than a single photo generation. This is because model variants are usually produced over several years. In the specific case of C3 I, productions span the 2002 - 2012 period, and the generated images for Prompt 2 should include both spansworn-out and brand new photos. This could be implemented by training an intermediate module to analyze the initial prompts to extract important time-related aspects and incorporate them in reformulated prompt variants to produce diversified images. In practice, the success of such an approach would depend on the annotation granularity for training images. CaMiT could be extended with more refined image captions to handle more refined generation criteria, but such work falls outside the immediate scope.
>
> 2.Data Imbalance: We intentionally preserved CaMiT’s natural, heavy-tailed distribution to reflect real-world availability across years. The distance-based classifiers used in TICL offer a simple yet effective mitigation of imbalance (see reference (Masana et al, 2022) and (Rebuffi et al, 2017)). To alleviate the heavy imbalance during training, future work could leverage our time-aware generation framework for targeted augmentation (synthesizing additional samples for sparse years or models) or explore loss-weighting in time-incremental settings, as proposed in (Jodelet et al, 2023).
>
> Masana, Marc, et al. "Class-incremental learning: survey and performance evaluation on image classification." IEEE Transactions on Pattern Analysis and Machine Intelligence 45.5 (2022): 5513-5533.
> Rebuffi, Sylvestre-Alvise, et al. "icarl: Incremental classifier and representation learning." Proceedings of the IEEE conference on Computer Vision and Pattern Recognition. 2017.
> Jodelet, Quentin, et al. "Class-incremental learning using diffusion model for distillation and replay." Proceedings of the IEEE/CVF International Conference on Computer Vision. 2023

---

> > ### Comment · Reviewer_eyUP · 2025-08-03
> >
> > Thank you for clarifying that CaMiT could be extended with more refined image captions to handle more refined generation criteria, which would support generating the same vehicle model with different time-aware appearances.
> >
> > I would like to further emphasize that a dataset capturing the **appearance change** of the same vehicle over time would be of greater value to the research community than one focusing on **design changes** across different model variants.
> > - The **appearance change** dataset can support tasks such as time-aware vehicle re-identification and generation with varying degrees of wear and tear. For example, training on images of the same 2002 C3 I between 2002 and 2012 with different degrees of wear would allow a generation model to extrapolate its 2015 appearance, which is most likely more worn out.
> > - However, the **design change** does not follow a continuous trajectory over time. For example, when a car company replaces its design team with a new one, the entire design direction might shift abruptly, making it less meaningful to learn. If the model is trained on various C3 I designs from 2002 to 2012, it is unlikely to generate/predict what a hypothetical 2015 version would look like, because the design evolution is hard to be learned in a causal sense.
> > - The current dataset uses the user uploaded timestamps for data labeling and hasn't been distinguished between design change and appearance change. This should be addressed.
> >
> > I appreciate the authors’ clarification and discussed future directions. I maintain my score and suggest accepting this paper.

---

> > ### Author Response · Authors · 2025-08-06
> >
> > We thank the reviewer for this valuable insight. We agree that modeling the appearance change of the same car over time would greatly enhance tasks like re-identification and progressive wear generation. We will add a paragraph in the Limitations noting that CaMiT currently mixes design changes and physical aging and that future works could disentangle appearance change and design change.

---

### Official Review · Reviewer_WmQS · 2025-07-03

**Rating:** 5
**Confidence:** 3

**Summary:**

CaMiT presents an image dataset consisting of various car models over the time span of 16 years. Each image is labelled with the car's brand, model, model year, image year and corresponding bounding boxes, providing detailed information of car models across history and a new dataset for the time-aware classification task. The data is sourced from Flickr, cropped and carefully filtered to only include individual samples with acceptable quality (size, occlusion, detectability). A combination of open source Qwen2.5-7B and closed GPT-4o annotates filtered samples then and only models verified through both models are kept for the main dataset. A separate larger pre-training dataset classified per image year is kept as well. An analysis of image features across dataset shows clear changes in visual features when models are upgraded with a significant design changes.
Secondly the authors present the usage of the dataset for static pretraining, time-incremental pretraining, and time-incremental classifier learning on large pre-trained models and a trained specialized car model. In addition to the descriptive tasks time-aware image generation experiments show year- and model-conditional generation results.

**Dataset Code Accessibility:**

Yes

**Dataset Code Comments:**

The README is clear and enables full reproducibility of the presented results.

**Ethical Comments:**

See paper limitations.

**Ethical Considerations:**

No, there are no or only very minor ethics concerns

**Final Justification:**

I thank the authors considering weaknesses raised on their initial submission and providing answers and additional experiments to questions from all reviewers! The answers provided in the rebuttal are sufficient for me and considering a polished figure and improved additional table on related work, that was promised by the authors, I don't see any reason to change my rating and suggest accepting this paper.

**Limitations Weaknesses:**

While this seems like an overall very mature paper, there are minor weaknesses:
- Minor complains about Figure 2: I think that the readability can be improved by clearly separating  labels of steps and descriptions as well as incorporating a little more details on individual steps outputs and effect on the dataset filtering.
- While mitigation techniques are presented and the authors acknowledge the cost and effort of manual labeling, it is still possible the specific samples and the dataset distribution could depend on the VLM models used to filter and annotate initial samples.
- A graphical comparison with other (time-incremental) datasets would be a nice addition to the related work section, which is generally dense in content and references.
- Additionally a more detailed comparison with TIC-DataComp would help to understande the advantages of this dataset better.

**Strengths Contributions:**

The overall quality of the paper, the supplementary and the code base in their writing and presentation is  extremely good and does not leave a lot of room for complains,and I want to highlight the following points:
- Nearly all figures and tables in the main paper and supplementary are well structured and self-explanatory
- The depth of explanation of all filters and the dataset sample selection, especially the complete explanation in the supplementary, are sound and seem to provide a lot of mitigation techniques against biases and low data quality samples.
- Fig. 1 is a great teaser and motivation for the dataset and why the datasets can be of value to the community.
- Even though the dataset seems to be a bit niche, I think the presented quality and the presented evaluation provide enough value to the community. It also provides data at a level of details and size that is of relevance.
- Experiments on pretraining and classification provide interesting insights and the observation of phenomena as forgetting in SPT are valuable insights on large pretrained models  (I am by far not an expert in classification).
- Additionally I want to highlight that the generation results are pretty convincing, even though this is a smaller part of the presented evaluation.

---

> ### Author Rebuttal · Authors · 2025-07-29
>
> Thank you for you thorough and constructive review, and for highlighting the clarity of our figures, the rigor of our filtering pipeline, and the value of our generation results.
>
> Below we address each of your suggestions.
>
> 1. Figure 2 readability: For more clarity, we will change Figure 2 in the camera-ready version by placing the step label outside the boxes. We will also describe the individual steps and their effects on the dataset filter in the Figure 2 caption.
>
> 2. Dependence on VLM models: We acknowledge that the semi-automatic pipeline, which includes VLMs, can potentially affect the dataset distribution, but to ensure the labeling reliability under reasonable cost, we can only resort to a semi-automatic pipeline. Moreover, across our dataset, we observe that major shifts in the year-by-year embedding distance matrix correspond closely to real-world introduction of new car variants, suggesting that our annotations capture genuine temporal dynamics.
>
> 3. Graphical comparison with other time-incremental datasets: We will add a compact table to compare the datasets that are covered in the related works section, including dimensions such as modality, size, label/caption granularity, domain, and temporal coverage.
>
> 4. More detailed comparison with TIC-DataComp: Our core question  is "How can fine-grained technological artifacts be modeled over long time spans ?" First, TIC-DataComp is much larger and covers a broad, mixed set of visual concepts, whereas CaMiT focuses specifically on cars, allowing for controlled fine-grained study. Second, CaMiT spans over 19 years compared to TIC-DataComp’s 10 years (and it only provides sufficient coverage for the 2017-2022 period, see figure 2 in the appendix), which better captures gradual evolution. Third, TIC-DataComp includes images and associated captions without manual verification, while CaMiT is annotated to ensure high-quality content and enable quantitative performance evaluation. We will discuss these differences in the revised version of the submission.

---

> > ### Comment · Reviewer_WmQS · 2025-08-02
> >
> > I thank the authors considering weaknesses raised on their initial submission and providing answers and additional experiments to questions from all reviewers! The answers provided in the rebuttal are sufficient for me and considering a polished figure and improved additional table on related work, that was promised by the authors, I don't see any reason to change my rating and suggest accepting this paper.

---

> > > ### Author Response · Authors · 2025-08-06
> > >
> > > Thank you for this positive comment regarding the rebuttal!

---

### Decision · Program_Chairs · 2025-09-18

**Decision:**

Accept (poster)

**Comment:**

The authors introduce a new dataset (CaMiT) for visual classification of different car models from images. It contains 800K images for 190 car models, in addition to 5M unlabelled images.

There were some limitations of the work, e.g., the images are obtained via an semi-automatic pipeline which may result in non trivial label error. However, in spite of this the reviewers were very supportive of the work and recommended that it should be accepted.

The authors are strongly encouraged to discuss the above limitations in the revised text in addition to the other comments from the reviewers.

Note, the paper flagged by kcf5 is not very relevant to this work, and thus it is not important to cite it in the final camera ready text.